# Epistasis between mutator alleles contributes to germline mutation spectrum variability in laboratory mice

Thomas A Sasani[1]*, Aaron R Quinlan[1,2], Kelley Harris[3,4]*

[1]Department of Human Genetics, University of Utah, Salt Lake City, United States; [2]Department of Biomedical Informatics, University of Utah, Salt Lake City, United States; [3]Department of Genome Sciences, University of Washington, Seattle, United States; [4]Herbold Computational Biology Program, Fred Hutch Cancer Center, Seattle, United States

*For correspondence:
thomas.a.sasani@gmail.com (TAS);
harriske@uw.edu (KH)

Competing interest: The authors declare that no competing interests exist.

**Abstract** Maintaining germline genome integrity is essential and enormously complex. Although many proteins are involved in DNA replication, proofreading, and repair, *mutator alleles* have largely eluded detection in mammals. DNA replication and repair proteins often recognize sequence motifs or excise lesions at specific nucleotides. Thus, we might expect that the spectrum of de novo mutations – the frequencies of C>T, A>G, etc. – will differ between genomes that harbor either a mutator or wild-type allele. Previously, we used quantitative trait locus mapping to discover candidate mutator alleles in the DNA repair gene *Mutyh* that increased the C>A germline mutation rate in a family of inbred mice known as the BXDs (Sasani et al., 2022, Ashbrook et al., 2021). In this study we developed a new method to detect alleles associated with mutation spectrum variation and applied it to mutation data from the BXDs. We discovered an additional C>A mutator locus on chromosome 6 that overlaps *Ogg1*, a DNA glycosylase involved in the same base-excision repair network as *Mutyh* (David et al., 2007). Its effect depends on the presence of a mutator allele near *Mutyh*, and BXDs with mutator alleles at both loci have greater numbers of C>A mutations than those with mutator alleles at either locus alone. Our new methods for analyzing mutation spectra reveal evidence of epistasis between germline mutator alleles and may be applicable to mutation data from humans and other model organisms.

## eLife assessment

By developing a novel method for detecting genetic variants associated with germline mutation spectrum variation, this **important** study identifies a new "mutator" locus in a population of inbred mouse strains, although the causal gene(s) and allele(s) within this locus remain uncertain. The authors further demonstrate that this new mutator locus interacts epistatically with a previously identified mutator allele on C>A mutation rate, showcasing the complexity of the genetic basis underlying variation in mutation rate and spectrum. Evidence for major findings in this paper is **convincing**, and the new method has the potential to be applicable to a variety of experimental systems and natural populations.

## Introduction

Germline mutation rates reflect the complex interplay between DNA proofreading and repair pathways, exogenous sources of DNA damage, and life-history traits. For example, parental age is an important determinant of mutation rate variability; in many mammalian species, the number of

germline de novo mutations observed in offspring increases as a function of paternal and maternal age (*Jónsson et al., 2017*; *Sasani et al., 2019*; *Wang et al., 2022*; *Wu et al., 2020*; *Lindsay et al., 2019*). Rates of germline mutation accumulation are also variable across human families (*Sasani et al., 2019*; *Rahbari et al., 2016*), likely due to either genetic variation or differences in environmental exposures. Although numerous protein-coding genes contribute to the maintenance of genome integrity, genetic variants that increase germline mutation rates, known as *mutator alleles*, have proven difficult to discover in mammals.

The dearth of observed germline mutators in mammalian genomes is not necessarily surprising, since alleles that lead to elevated germline mutation rates will likely have deleterious consequences and be purged by negative selection if their effect sizes are large (*Lynch et al., 2016*). Moreover, germline mutation rates are relatively low, and direct mutation rate measurements require whole-genome sequencing data from both parents and their offspring. As a result, large-scale association studies – which have been used to map the contributions of common genetic variants to many complex traits – are not currently well powered to investigate the polygenic architecture of germline mutation rates (*Kessler et al., 2020*).

Despite these challenges, less traditional strategies have been used to identify a small number of mutator alleles in humans, macaques (*Stendahl et al., 2023*), and mice. By focusing on families with rare genetic diseases, a recent study discovered two mutator alleles that led to significantly elevated rates of de novo germline mutation in human genomes (*Kaplanis et al., 2022*). Other groups have observed mutator phenotypes in the germlines and somatic tissues of adults who carry cancer-predisposing inherited mutations in the POLE/POLD1 exonucleases (*Robinson et al., 2021*; *Sherwood et al., 2023*). Candidate mutator loci were also found by identifying human haplotypes from the Thousand Genomes Project with excess counts of derived alleles in genomic windows (*Seoighe and Scally, 2017*).

In mice, a germline mutator allele was recently discovered by sequencing a large family of inbred mice (*Sasani et al., 2022*). Commonly known as the BXDs, these recombinant inbred lines (RILs) were derived from either F2 or advanced intercrosses of C57BL/6J and DBA/2J, two laboratory strains that exhibit significant differences in their germline mutation spectra (*Ashbrook et al., 2021*; *Dumont, 2019*). At the time of their whole-genome sequencing in 2017, the BXDs had been maintained via brother-sister mating for up to 180 generations, and each BXD had therefore accumulated hundreds or thousands of germline mutations on a nearly homozygous linear mosaic of parental *B* and *D* haplotypes. Due to their husbandry in a controlled laboratory setting, the BXDs are largely free from confounding by environmental heterogeneity, and the effects of selection on de novo mutations have been attenuated by strict inbreeding (*Halligan and Keightley, 2009*).

In this previous study, whole-genome sequencing data from the BXD family were used to map a quantitative trait locus (QTL) for the C>A mutation rate (*Sasani et al., 2022*). Germline C>A mutation rates were nearly 50% higher in mice with *D* haplotypes at the QTL, likely due to genetic variation in the DNA glycosylase *Mutyh* that reduced the efficacy of oxidative DNA damage repair. Pathogenic variants of *Mutyh* also appear to act as mutators in human germline and somatic tissues (*Sherwood et al., 2023*; *Robinson et al., 2022*). Importantly, the QTL did not reach genome-wide significance in a scan for variation in overall germline mutation rates, which were only modestly higher in BXDs with *D* alleles, demonstrating the utility of mutation spectrum analysis for mutator allele discovery. Close examination of the mutation spectrum is likely to be broadly useful for detecting mutator alleles, as genes involved in DNA proofreading and repair often recognize particular sequence motifs or excise specific types of DNA lesions (*Carlson et al., 2020*). Mutation spectra are usually defined in terms of $k$-mer nucleotide context; the 1-mer mutation spectrum, for example, consists of six mutation types after collapsing by strand complement (C>T, C>A, C>G, A>T, A>C, A>G), while the 3-mer mutation spectrum contains 96 (each of the 1-mer mutations partitioned by trinucleotide context).

Although mutation spectrum analysis can enable the discovery of mutator alleles that affect the rates of specific mutation types, early implementations of this strategy have suffered from a few drawbacks. For example, performing association tests on the rates or fractions of every $k$-mer mutation type can quickly incur a substantial multiple testing burden. Since germline mutation rates are generally quite low, estimates of $k$-mer mutation type frequencies from individual samples can also be noisy and imprecise. In populations of RILs, inbreeding duration can also vary substantially; for example, some BXDs were inbred for only 20 generations, while others were inbred for nearly 200. As a result,

the variance of individual $k$-mer mutation rate estimates in those populations will be higher than if all samples were inbred for the same duration. We were therefore motivated to develop a statistical method that could overcome the sparsity of de novo mutation spectra, eliminate the need to test each $k$-mer mutation type separately, and enable sensitive detection of alleles that influence the germline mutation spectrum.

Here, we present a new mutation spectrum association test, called 'aggregate mutation spectrum distance' (AMSD), that minimizes multiple testing burdens and mitigates the challenges of sparsity in de novo mutation datasets. We leverage this method to re-analyze germline mutation data from the BXD family and find compelling evidence for a second mutator allele that was not detected using previous approaches. The new allele appears to interact epistatically with the mutator that was previously discovered in the BXDs, further augmenting the C>A germline mutation rate in a subset of inbred mice. Our observation of epistasis suggests that mild DNA repair deficiencies can compound one another, as mutator alleles chip away at the redundant systems that collectively maintain germline integrity.

## Results

### A novel method for detecting mutator alleles

We developed a statistical method, termed the 'AMSD', to detect loci that are associated with mutation spectrum variation in RILs (*Figure 1*; Materials and methods). Our approach leverages the fact that mutator alleles often leave behind distinct and detectable impressions on the *mutation spectrum*, even if they increase the overall mutation rate by a relatively small amount. Given a population of haplotypes, we assume that each has been genotyped at the same collection of biallelic loci and that each harbors de novo mutations which have been partitioned by $k$-mer context (*Figure 1*). At every locus, we calculate a cosine distance between the aggregate mutation spectra of the sets of haplotypes that inherited each parental allele. Using permutation tests, we then identify genetic markers whose reference and alternate alleles are associated with aggregate mutation spectra that are more distinct than what we'd expect by random chance. To account for polygenic effects on the mutation process that might be shared between BXDs, we also regress the mutation spectrum cosine distance at each marker against the genetic similarity between haplotype groups, and assess significance using the fitted residuals (which we call the 'adjusted' cosine distances) (Materials and methods).

Using simulated data, we find that our method's power is primarily limited by the initial mutation rate of the $k$-mer mutation type affected by a mutator allele and the total number of de novo mutations used to detect it (*Figure 1—figure supplement 1*). Given 100 haplotypes with an average of 500 de novo germline mutations each, AMSD has approximately 90% power to detect a mutator allele that increases the C>A de novo mutation rate by as little as 20%. However, the approach has less than 20% power to detect a mutator of identical effect size that augments the C>G mutation rate, since C>G mutations are expected to make up a smaller fraction of all de novo germline mutations to begin with. Simulations also demonstrate that our approach is well powered to detect large-effect mutator alleles (e.g. those that increase the mutation rate of a specific $k$-mer by 50%), even with a relatively small number of mutations per haplotype (*Figure 1—figure supplement 1*). Both AMSD and traditional QTL mapping have similar power to detect alleles that augment the rates of individual 1-mer mutation types (*Figure 1—figure supplement 2*), but AMSD has a number of potential advantages for mutator allele discovery. For example, we find that AMSD is better powered than QTL mapping when the number of simulated de novo mutations is allowed to vary (by a factor of 20) across haplotypes (*Figure 1—figure supplement 3*) and when mutator allele frequencies are less than 50% (*Figure 1—figure supplement 4*). However, we also caution that many of the parameters used in our simulations are specific to the BXD mice (e.g. numbers of haplotypes, average numbers of mutations, expected allele frequencies at markers), and do not necessarily reflect the power of AMSD in other populations.

### Re-identifying a mutator allele on chromosome 4 in the BXDs

We applied our AMSD method to 117 BXDs (Materials and methods) with a total of 65,552 de novo germline mutations (*Sasani et al., 2022*). Using mutation data that were partitioned by 1-mer nucleotide context, we discovered a locus on chromosome 4 that was significantly associated with mutation

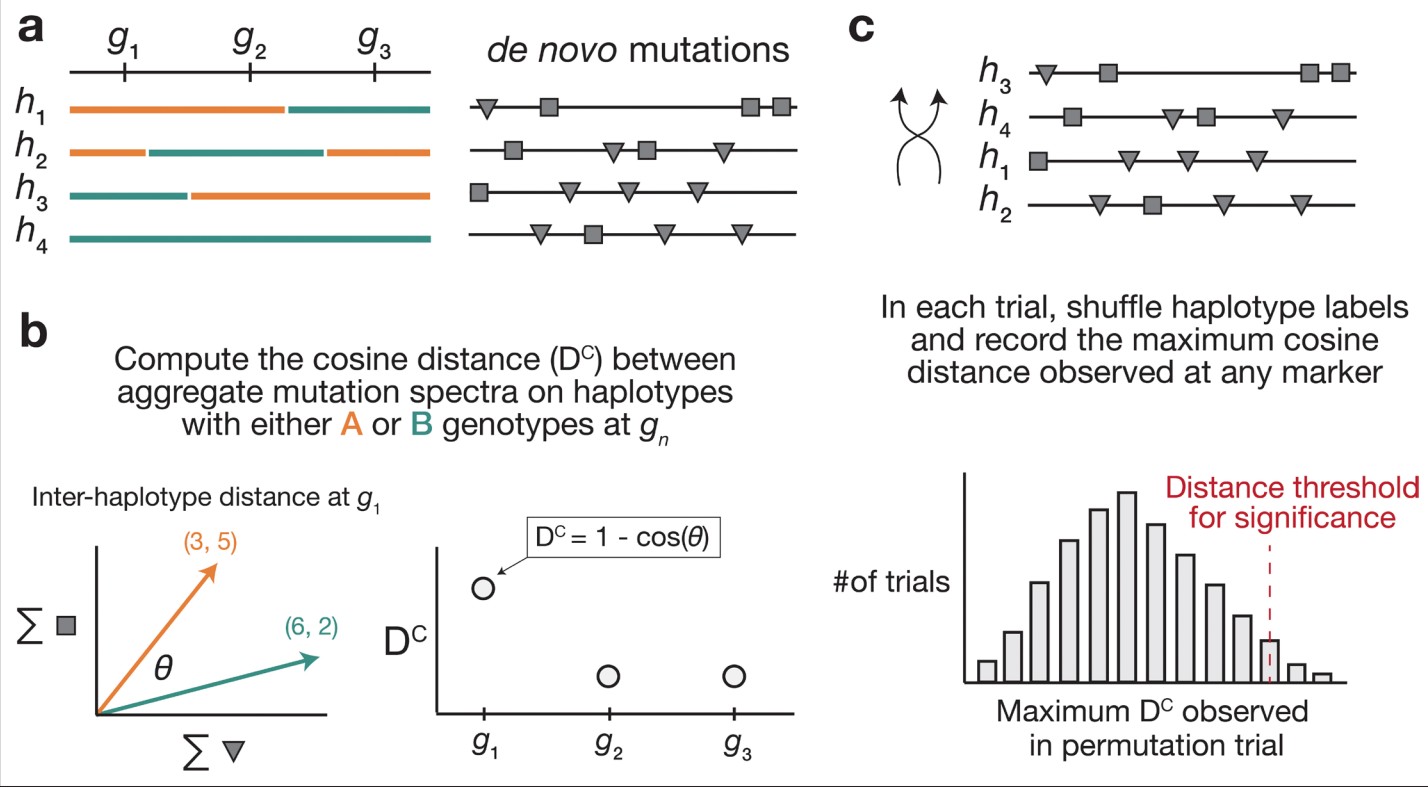

**Figure 1.** Overview of aggregate mutation spectrum distance method for discovering mutator alleles. (**a**) A population of four haplotypes has been genotyped at three informative markers ($g_1$ through $g_3$); each haplotype also harbors unique de novo germline mutations. In practice, de novo mutations are partitioned by $k$-mer context; for simplicity in this toy example, de novo mutations are simply classified into two possible mutation types (gray squares represent C>(A/T/G) mutations, while gray triangles represent A>(C/T/G) mutations). (**b**) At each informative marker $g_n$, we calculate the total number of each mutation type observed on haplotypes that carry each parental allele (i.e. the aggregate mutation spectrum) using all genome-wide de novo mutations. For example, haplotypes with A (orange) genotypes at $g_1$ carry a total of three 'triangle' mutations and five 'square' mutations, and haplotypes with B (green) genotypes carry a total of six triangle and two square mutations. We then calculate the cosine distance between the two aggregate mutation spectra, which we call the 'aggregate mutation spectrum distance'. Cosine distance can be defined as $1 - cos(\theta)$, where $\theta$ is the angle between two vectors; in this case, the two vectors are the two aggregate spectra. We repeat this process for every informative marker $g_n$. (**c**) To assess the significance of any distance peaks in (b), we perform permutation tests. In each of $N$ permutations, we shuffle the haplotype labels associated with the de novo mutation data, run a genome-wide distance scan, and record the maximum cosine distance encountered at any locus in the scan. Finally, we calculate the $1 - p$ percentile of the distribution of those maximum distances to obtain a genome-wide cosine distance threshold at the specified value of $p$.

The online version of this article includes the following figure supplement(s) for figure 1:

**Figure supplement 1.** Simulations to assess the power of the aggregate mutation spectrum distance method.

**Figure supplement 2.** Comparing power between the aggregate mutation spectrum distance (AMSD) method and QTL mapping.

**Figure supplement 3.** Comparing power between the aggregate mutation spectrum distance method (AMSD) and QTL mapping with variable counts of simulated mutations.

**Figure supplement 4.** Comparing power between the aggregate mutation spectrum distance method (AMSD) and QTL mapping with variable mutator allele frequencies.

spectrum variation (**Figure 2a**; maximum adjusted cosine distance of 1.20e-2 at marker ID rs27509845; position 118.28 Mbp in GRCm38/mm10 coordinates; 90% bootstrap confidence interval from 114.79 to 118.75 Mbp).

Using QTL mapping, we previously identified a nearly identical locus on chromosome 4 that was significantly associated with the C>A germline mutation rate in the BXDs (**Sasani et al., 2022**). This locus overlapped 21 protein-coding genes that were annotated by the Gene Ontology (GO) as being involved in 'DNA repair', but only one of those genes contained nonsynonymous differences between the two parental strains: *Mutyh*. *Mutyh* encodes a protein involved in the base-excision repair (BER) of 8-oxoguanine (8-oxoG), a DNA lesion caused by oxidative damage, and prevents the accumulation of

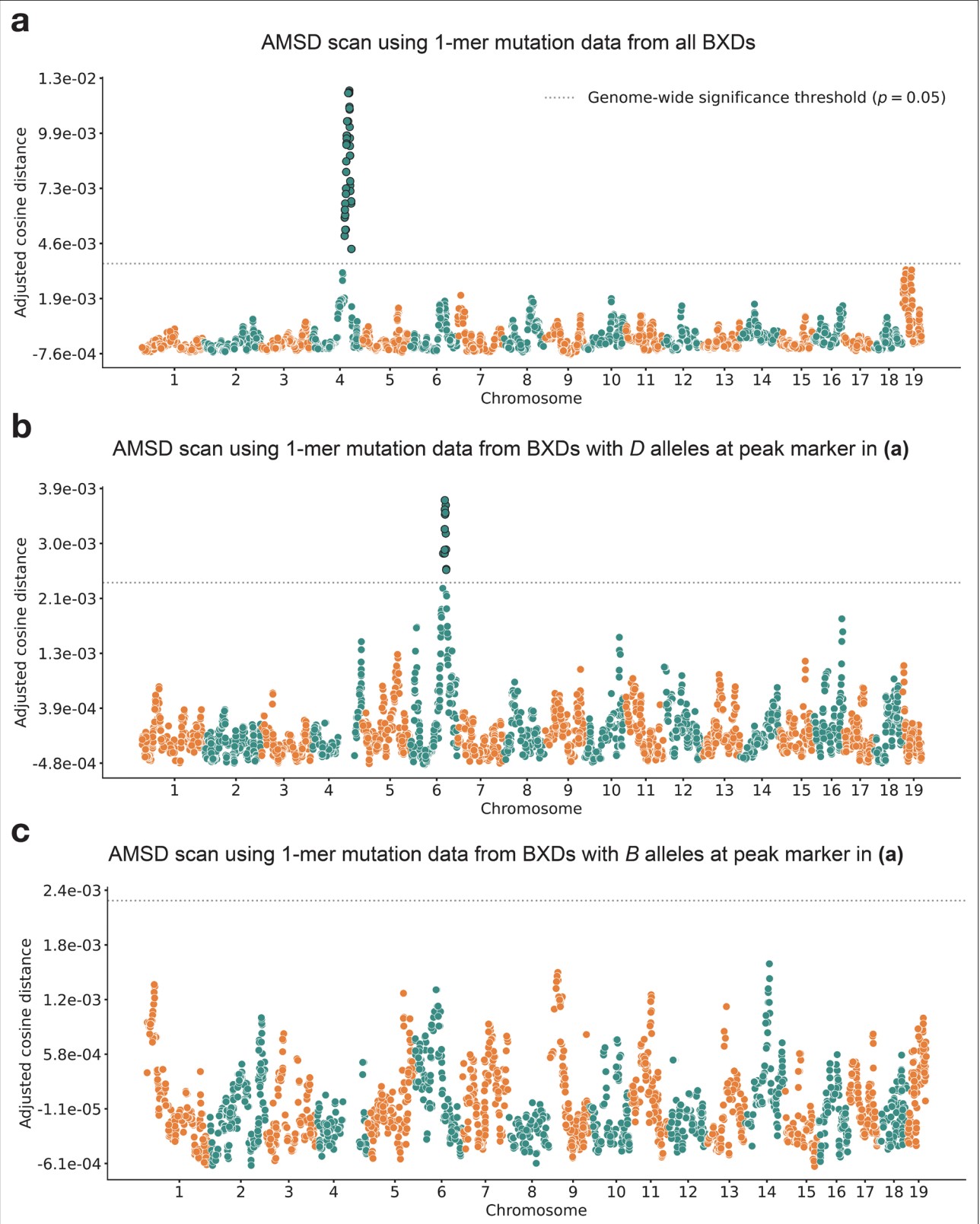

**Figure 2.** Results of aggregate mutation spectrum distance scans in the BXDs. **(a)** Adjusted cosine distances between aggregate 1-mer de novo mutation spectra of BXD strains ($n$ = 117 strains; 65,552 total mutations) with either D or B alleles at 7128 informative markers. Cosine distance threshold at p=0.05 was calculated by performing 10,000 permutations of the BXD mutation data, and is shown as a dotted gray line. **(b)** Adjusted cosine distances between aggregate 1-mer de novo mutation spectra of BXD strains with D alleles at rs27509845 ($n$ = 66 strains; 42,171 total mutations) and

*Figure 2 continued on next page*

Figure 2 continued

either D or B alleles at 6957 informative markers. Cosine distance threshold at p = 0.05 was calculated by performing 10,000 permutations of the BXD mutation data, and is shown as a dotted gray line. (**c**) Adjusted cosine distances between aggregate 1-mer de novo mutation spectra on BXD strains with B alleles at rs27509845 (n = 44 strains; 22,645 total mutations) and either D or B alleles at 6957 informative markers. Cosine distance threshold at p = 0.05 was calculated by performing 10,000 permutations of the BXD mutation data, and is shown as a dotted gray line.

The online version of this article includes the following figure supplement(s) for figure 2:

**Figure supplement 1.** Quantitative trait locus (QTL) scans for mutation spectrum phenotypes.

---

C>A mutations (*David et al., 2007*; *Viel et al., 2017*; *Pilati et al., 2017*). C>A germline mutation fractions are nearly 50% higher in BXDs that inherit *D* genotypes at marker ID rs27509845 (the marker at which we observed the highest adjusted cosine distance on chromosome 4) than in those that inherit *B* genotypes (*Figure 3*, *Sasani et al., 2022*).

## An additional germline mutator allele on chromosome 6

After confirming that AMSD could recover the mutator locus overlapping *Mutyh*, we tested its ability to identify additional mutator loci in the BXDs. To eliminate potential confounding of the mutation spectrum landscape by the large-effect mutator locus on chromosome 4, we performed AMSD scans that were conditional on the presence of either *D* or *B* alleles at rs27509845. We hypothesized that such conditioning might reveal epistatic interactions between alleles at the chromosome 4 locus and mutator alleles elsewhere in the genome. Specifically, we divided the BXDs into those with either *D* (n = 66) or *B* (n = 44) genotypes at rs27509845 (n = 7 BXDs were heterozygous) and ran an AMSD scan using each group separately (*Figure 2b–c*). We excluded the BXD68 RIL from these scans, since we previously found that BXD68 harbors a strain-private C>A mutator allele of even larger effect than that of a *D* allele at rs27509845 (*Sasani et al., 2022*).

Using the BXDs with *D* genotypes at rs27509845, we identified a locus on chromosome 6 that was significantly associated with mutation spectrum variation (*Figure 2b*; maximum adjusted cosine distance of 3.69e-3 at marker rs46276051; position 111.27 Mbp in GRCm38/mm10 coordinates; 90% bootstrap confidence interval from 95.01 to 114.02 Mbp). This signal was specific to BXDs with *D* genotypes at the rs27509845 locus, as we did not observe any new mutator loci after performing an AMSD scan using BXDs with *B* genotypes at rs27509845 (*Figure 2c*). The peak markers on chromosomes 4 and 6 did not exhibit strong linkage disequilibrium ($R^2$ = 4e-5). We also performed QTL scans for the fractions of each 1-mer mutation type using the same mutation data, but none produced a genome-wide significant log-odds (LOD) score at any locus (*Figure 2—figure supplement 1*; Materials and methods).

We queried the region surrounding the top marker on chromosome 6 (± the 90% bootstrap confidence interval) and discovered 64 protein-coding genes, of which 4 were annotated as being related to DNA repair in the Gene Ontology (GO) term classification system (*Ashburner et al., 2000*; *Gene Ontology Consortium, 2021*). These four genes are *Fancd2*, *Ogg1*, *Setmar*, and *Rad18*. None of the remaining genes were annotated with a cellular function that would obviously contribute to a germline mutator phenotype; however, many of these GO annotations are imperfect and/or incomplete. Although we focus our analysis on DNA repair genes, it remains possible that other genes within the confidence interval might underlie the C>A mutator phenotype we identified in the BXDs.

Of the annotated DNA repair genes within the confidence interval, two (*Ogg1* and *Setmar*) harbor nonsynonymous differences between the parental C57BL/6J and DBA/2J strains (*Table 1*). *Ogg1* encodes a key member of the BER response to oxidative DNA damage (a pathway that also includes *Mutyh*), and in mice *Setmar* encodes a SET domain-containing histone methyltransferase; both *Ogg1* and *Setmar* are expressed in mouse gonadal cells. Because the bootstrap can exhibit poor coverage in QTL mapping studies (*Manichaikul et al., 2006*), we also scanned an interval ±5 Mbp from the peak AMSD marker on chromosome 6 for additional candidate genes. Although the choice of a 10 Mbp interval is somewhat arbitrary, the interval does contain an additional plausible candidate: *Mbd4*, a protein-coding gene involved in BER that also harbors a nonsynonymous difference between the BXD parental strains and is expressed in mouse gonads (*Table 1*).

We also considered the possibility that expression quantitative trait loci (eQTLs), rather than nonsynonymous mutations, could contribute to the C>A mutator phenotype associated with the locus on chromosome 6. Using GeneNetwork (*Mulligan et al., 2017*) we mapped eQTLs for the

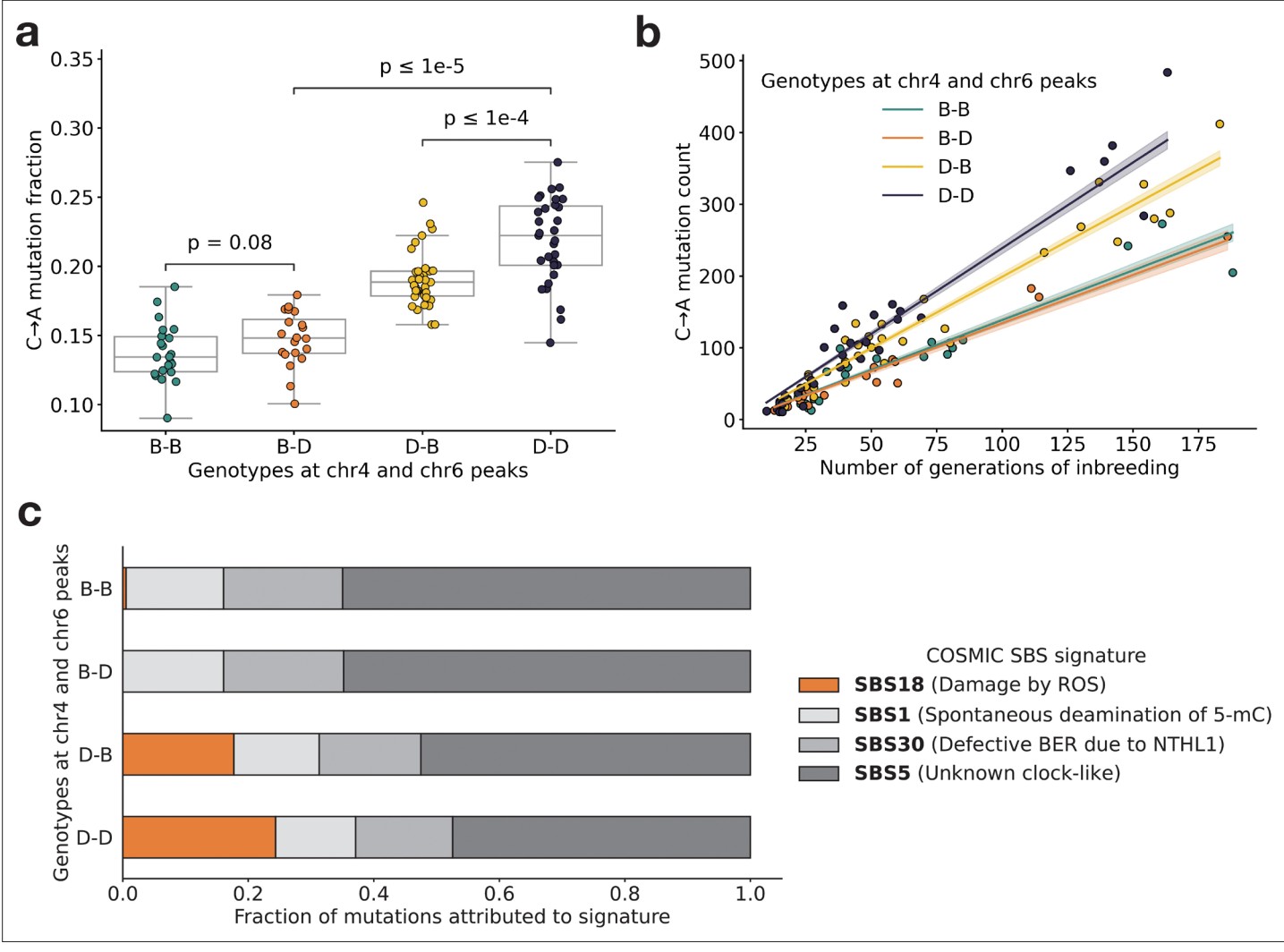

**Figure 3.** BXD mutation spectra are affected by alleles at both mutator loci. (**a**) C>A de novo germline mutation fractions in BXDs with either D or B genotypes at markers rs27509845 (chr4 peak) and rs46276051 (chr6 peak). Distributions of C>A mutation fractions were compared with two-sided Mann-Whitney U-tests; annotated p-values are uncorrected. B-B vs. B-D comparison: U-statistic = 149.0, p = 7.58e-2; B-D vs D-D comparison: U-statistic = 21.0, p = 2.61e-8; D-B vs D-D comparison: U-statistic = 232.5, p = 6.99e-5. (**b**) The count of C>A de novo germline mutations in each BXD plotted against the number of generations for which it was inbred. Lines represent predicted C>A counts in each haplotype group from a generalized linear model (Poisson family, identity link), and shading around each line represents the 95% confidence interval. (**c**) Germline mutations in each BXD were assigned to COSMIC single-base substitution (SBS) mutation signatures using SigProfilerExtractor (*Islam et al., 2022*). After grouping BXDs by their genotypes at rs27509845 and rs46276051, we calculated the fraction of mutations in each group that was attributed to each signature. The proposed etiologies of each mutation signature are: SBS1 (spontaneous deamination of methylated cytosine nucleotides at CpG contexts), SBS5 (unknown, clock-like signature), SBS18 (damage by reactive oxygen species, related to SBS36 and defective base-excision repair due to loss-of-function mutations in MUTYH), and SBS30 (defective base-excision repair due to NTHL1 mutations).

The online version of this article includes the following figure supplement(s) for figure 3:

**Figure supplement 1.** Mutation spectra comparison in BXD strains.

**Figure supplement 2.** Mutation spectra comparison in Sanger Mouse Genomes Project (MGP) strains.

**Figure supplement 3.** Frequency of nonsynonymous DNA repair mutations in wild mice.

four aforementioned DNA repair genes (as well as *Mbd4*) in a number of tissues, though we did not have access to expression data from germline cells. Notably, *D* alleles near the cosine distance peak on chromosome 6 were significantly associated with decreased *Ogg1* expression in kidney, liver, hippocampus, and gastrointestinal tissues (*Supplementary file 1*). Although these cis-eQTLs are challenging to interpret (given their tissue specificity and our lack of access to germline expression

**Table 1.** Nonsynonymous mutations in DNA repair genes near the chr6 peak.

| Gene name | Ensembl transcript name | Nucleotide change | Amino acid change | Position in GRCm38/ mm10 coordinates | PhyloP conservation score | SIFT prediction |
|---|---|---|---|---|---|---|
| *Setmar* | ENSMUST00000049246 | C>T | p.Leu103Phe | chr6:108,075,853 | 0.422 | 0.0 (intolerant/deleterious) |
| *Setmar* | ENSMUST00000049246 | T>G | p.Ser273Arg | chr6:108,076,365 | –0.355 | 0.3 (tolerant/benign) |
| *Ogg1* | ENSMUST00000032406 | A>G | p.Thr95Ala | chr6:113,328,510 | –0.016 | 0.84 (tolerant/benign) |
| *Mbd4* | ENSMUST00000032469 | C>T | p.Asp129Asn | chr6:115,849,644 | 2.28 | 0.02 (intolerant/deleterious) |

data), the presence of strong-effect cis-eQTLs for *Ogg1* suggests that the C>A mutator phenotype observed in the BXDs may be mediated by regulatory, rather than protein-altering, variants.

Finally, we queried a dataset of structural variants (SVs) identified via high-quality, long-read assembly of inbred laboratory mouse strains (*Ferraj et al., 2023*) and found 176 large insertions or deletions (>100 bp) within the 90% bootstrap confidence interval around the cosine distance peak on chromosome 6; none overlapped the exonic sequences of protein-coding genes. One protein-coding gene involved in DNA repair (*Rad18*) harbored an intronic deletion within the interval on chromosome 6 (chr6:112,629,618–112,636,619); however, additional experimental evidence will be needed to probe the functional impact of this SV.

### Evidence of epistasis between germline mutator alleles

Next, we more precisely characterized the effects of the chromosome 4 and 6 mutator alleles on mutation spectra in the BXDs. To pinpoint the mutation type(s) underlying the significant cosine distance peak on chromosome 6, we compared the aggregate counts of each 1-mer mutation type (plus CpG>TpG) on BXD haplotypes with *D* genotypes at `rs27509845` and either *D* or *B* genotypes at `rs46276051`. We found that C>A mutations were significantly enriched on BXD haplotypes with *D* genotypes at the chromosome 6 mutator locus, relative to those with *B* genotypes ($\chi^2$ statistic = 85.36, p = 2.48e-20). On average, C>A germline mutation fractions were significantly higher in BXDs with *D* alleles at both mutator loci than in BXDs with *D* alleles at either locus alone (*Figure 3a* and *Figure 3—figure supplement 1*). Among BXDs with *B* alleles at the locus overlapping *Mutyh*, those with *D* alleles on chromosome 6 did not exhibit significantly elevated C>A mutation fractions (*Figure 3a*). After controlling for inbreeding duration, we observed that C>A de novo mutation counts were always highest in BXDs with *D* alleles at both mutator loci (*Figure 3b*). After 100 generations of inbreeding, BXDs with *D* alleles at both mutator loci were predicted to have 238.9 C>A mutations (95% CI: 231.4–246.4), about 20% more than the 199.0 mutations (95% CI: 193.3–204.7) predicted in those with *D* and *B* alleles at the chromosome 4 and chromosome 6 loci, respectively (*Figure 3b*).

We also used SigProfilerExtractor (*Islam et al., 2022*) to assign the germline mutations in each BXD to single-base substitution (SBS) mutation signatures from the COSMIC catalog (*Tate et al., 2019*). Mutation signatures often reflect specific exogenous or endogenous sources of DNA damage, and the proportions of mutations attributable to particular SBS signatures can suggest a genetic or environmental etiology. The SBS1, SBS5, and SBS30 mutation signatures were active in nearly all BXDs, regardless of genotypes at the chromosome 4 and 6 mutator loci (*Figure 3c*). However, the SBS18 signature, which is dominated by C>A mutations and likely reflects unrepaired DNA damage from reactive oxygen species (ROS), was almost exclusively active in mice with *D* alleles at the chromosome 4 locus; the highest SBS18 activity was observed in mice with *D* alleles at both mutator loci (*Figure 3c*). SBS18 activity was lowest in mice with *D* alleles at the chromosome 6 mutator locus alone (*Figure 3c*), further demonstrating that *D* alleles at this locus are not sufficient to cause a mutator phenotype.

To more formally test for statistical epistasis, we fit a generalized (Poisson) linear model predicting counts of C>A mutations in each BXD as a function of genotypes at `rs27509845` and `rs46276051` (the markers with the largest adjusted cosine distance at the two mutator loci); the model also accounted for differences in inbreeding duration and sequencing coverage between the BXDs (Materials and methods). A model that included an interaction term between genotypes at the two markers fits the data significantly better than a model including only additive effects (p = 7.92e-7; Materials

and methods), indicating that the combined effects of *D* genotypes at both loci exceeded the sum of marginal effects of *D* genotypes at either locus alone.

To explore the effects of the two mutator loci in other inbred laboratory mice, we also compared the germline mutation spectra of Sanger Mouse Genomes Project (MGP) strains (*Keane et al., 2011*). *Dumont, 2019*, previously identified germline mutations that were private to each of the 29 MGP strains; these private variants likely represent recent de novo mutations (*Figure 3—figure supplement 2*). Only two of the MGP strains possess *D* genotypes at both the chromosome 4 and chromosome 6 mutator loci: DBA/1J and DBA/2J. As before, we tested for epistasis in the MGP strains by fitting two linear models predicting C>A mutation counts as a function of genotypes at the two mutator loci. A model incorporating an interaction term did not fit the MGP data significantly better than a model with additive effects alone (p = 0.806), so we are unable to confirm the signal of epistasis; however, this may be due to the smaller number of MGP strains with de novo germline mutation data.

## Some of the candidate mutator alleles are segregating in wild mice

To determine whether the candidate mutator alleles on chromosome 6 were segregating in natural populations, we queried previously published sequencing data generated from 67 wild-derived mice (*Harr et al., 2016*). Although these data are not a comprehensive sampling of the genetic diversity in wild mice, they include three subspecies of *M. musculus*, as well as the outgroup *M. spretus*. We found that the *Ogg1 D* allele was segregating at an allele frequency of 0.259 in *M. musculus domesticus*, the species from which C57BL/6J and DBA/2J derive the majority of their genomes (*Yang et al., 2007*), and was fixed in *M. musculus musculus*, *M. musculus castaneus*, and the outgroup *M. spretus* (*Figure 3—figure supplement 3*). The *Setmar* p.Ser273Arg *D* allele was also present at an allele frequency of 0.37 in *M. musculus domesticus*, while *D* alleles at the *Setmar* p.Leu103Phe variant were not observed in any wild *M. musculus domesticus* animals. *D* alleles at the *Mbd4* p.Asp129Asn variant were also absent from all wild mouse populations (*Figure 3—figure supplement 3*).

## Discussion

### Epistasis between germline mutator alleles

We have identified a locus on chromosome 6 that amplifies a C>A germline mutator phenotype in the BXDs, a family of inbred mice derived from the laboratory strains DBA/2J and C57BL/6J. DBA/2J (*D*) alleles at this locus have no significant effect on C>A mutation rates in mice that also harbor 'wild-type' C57BL/6J (*B*) alleles at a previously discovered mutator locus on chromosome 4 (*Sasani et al., 2022*). However, mice with *D* alleles at *both* loci have even higher mutation rates than those with *D* alleles at the chromosome 4 mutator locus alone (*Figure 3*). Epistatic interactions between mutator alleles have been previously documented in yeast (*Heck et al., 2006*) and in human cell lines (*Petljak et al., 2022*), but never to our knowledge in a whole-animal context.

Importantly, we discovered epistasis between germline mutator alleles in an unnatural population of model organisms that have been inbred by brother-sister mating in a highly controlled laboratory environment (*Ashbrook et al., 2021*). This breeding setup has likely attenuated the effects of natural selection on all but the most deleterious alleles (*Halligan and Keightley, 2009*), and may have facilitated the fixation of large-effect mutator alleles that would be less common in wild mice. Without fine-mapping the chromosome 6 mutator allele, however, we are unable to trace its origin to either a captive breeding colony of laboratory mice or a wild, outbreeding *M. musculus* population. If the mutator allele on chromosome 6 has even a weak deleterious fitness effect, there might be a greater likelihood that it persisted in captivity. Indeed, if purifying selection is required to keep mutation rates low, mutational pressure might cause mutation rates to rise in just a few generations of relaxed selection. This dynamic may explain the recent discovery of a large-effect mutator allele in a rhesus macaque research colony (*Stendahl et al., 2023*), as well as the observation that domesticated animals tend to have higher mutation rates than those in the wild (*Bergeron et al., 2023*). Although we have not conclusively fine-mapped the chromosome 6 mutator locus to a causal variant, we argue that nonsynonymous or regulatory variants in the DNA glycosylase *Ogg1* are the best candidates.

## Protein-coding genes that may underlie the chromosome 6 mutator locus

Five protein-coding genes involved in DNA repair overlap the C>A mutator locus on chromosome 6: *Ogg1*, a glycosylase that excises the oxidative DNA lesion 8-oxoG (*David et al., 2007*), *Setmar*, a histone methyltransferase involved in non-homologous end joining (NHEJ) of double-stranded breaks (DSBs) (*Fnu et al., 2011*; *Lee et al., 2005*), *Fancd2*, and *Rad18*. One other DNA repair gene, *Mbd4*, lies just outside of the 90% bootstrap confidence interval on chromosome 6 (but within a 10 Mbp interval around the peak AMSD marker). We are unable to conclusively determine that any of these genes harbors a causal variant underlying the observed C>A mutator phenotype, but several clues point to *Ogg1* as the most plausible candidate: *Ogg1* is a member of the same BER pathway as *Mutyh* (the gene that likely underlies the chromosome 4 mutator locus), contains a nonsynonymous fixed difference between the C57BL/6J and DBA/2J parental strains, and appears to be regulated by cis-eQTLs across a number of tissues within the BXD cohort.

The C57BL/6J and DBA/2J *Setmar* coding sequences differ by two missense variants (*Table 1*), one of which is predicted to be deleterious by in silico tools. The primate *SETMAR* ortholog is involved in NHEJ of DSBs, but its role in DNA repair appears to depend on the function of both a SET methyltransferase domain and a *Mariner* family transposase domain (*Lee et al., 2005*; *Kim et al., 2014*; *Beck et al., 2011*). Since the murine *Setmar* ortholog lacks the latter element, and because primate *SETMAR* is involved in a DNA repair process that is not expected to affect the rate of C>A mutations, this gene seems a priori unlikely to underlie the epistatic interaction between the chromosome 4 and 6 mutator loci in the BXDs (Appendix 1). Moreover, we did not observe any significant cis-eQTLs for *Setmar* across a variety of tissues in the BXD cohort (*Supplementary file 1*). None of the remaining DNA repair genes (*Fancd2* or *Rad18*) contains a nonsynonymous fixed difference between the C57BL/6J and DBA/2J parental strains, and none appear to be regulated by cis-eQTLs that would feasibly lead to a germline C>A mutator phenotype (*Supplementary file 1*); the only significant cis-eQTLs we observed for genes other than *Ogg1* were those at which *D* alleles actually led to *increased* expression.

## An *Ogg1* mutator allele might impair the excision of 8-oxoG lesions

*Ogg1* is a member of the same BER pathway as *Mutyh*, the protein-coding gene we previously implicated as harboring mutator alleles at the locus on chromosome 4 (*David et al., 2007*). Each of these genes has a distinct role in the BER response to oxidative DNA damage, and thereby the prevention of C>A mutations (*Markkanen, 2017*; *Ohno et al., 2014*). Following damage by ROS, *Ogg1* is able to recognize and remove 8-oxoG lesions that are base-paired with cytosine nucleotides; once 8-oxoG is excised, other members of the BER pathway are mobilized to restore a proper G:C base pair at the site. If an 8-oxoG lesion is not removed before the cell enters S-phase, adenine can be mis-incorporated opposite 8-oxoG during DNA replication (*Markkanen, 2017*). If this occurs, *Mutyh* can excise the mispaired adenine, leaving a one-nucleotide gap that is processed and filled with a cytosine by other BER proteins. The resulting C:8-oxoG base pair can then be 'returned' to *Ogg1* for excision and repair. Defects in the BER response to oxidative damage lead to significantly elevated rates of C>A mutation. For example, triple-knockout (KO) mice lacking *Ogg1*, *Mutyh*, and *Mth1* (which encodes an enzyme that prevents 8-oxo-dGTP from being incorporated during DNA synthesis; *Sakumi et al., 1993*) accumulate a 100-fold excess of 8-oxoG in their gonadal cells (*Ohno et al., 2014*). Almost 99% of de novo germline mutations in the *Ogg1/Mutyh/Mth1* triple KO mice are C>A transversions, demonstrating the clear role of 8-oxoG repair in preventing C>A mutation. Additionally, missense mutations and loss-of-heterozygosity in *Ogg1* have been associated with increased risk of human cancer (*Mahjabeen et al., 2012*; *Chevillard et al., 1998*), and copy-number losses of either *Ogg1* or *Mutyh* are linked to elevated rates of spontaneous C>A mutation in human neuroblastoma (*van den Boogaard et al., 2021*).

### Nonsynonymous mutations may underlie the chromosome 6 mutator phenotype

The p.Thr95Ala *Ogg1* missense variant is not predicted to be deleterious by the in silico tool SIFT (*Ng and Henikoff, 2003*), and occurs at a nucleotide that is not particularly well conserved across mammalian species (*Table 1*). We also observe that the *D* allele at p.Thr95Ala is segregating at an allele

frequency of approximately 26% among wild-derived *M. musculus domesticus* animals, and is fixed in other wild populations of *M. musculus musculus*, *M. musculus castaneus*, and *M. spretus*. Although we would expect a priori that *Ogg1* deficiency should lead to increased 8-oxoG accumulation and elevated C>A mutation rates, these lines of evidence suggest that p.Thr95Ala is not highly deleterious on its own, and might only exert a detectable effect on the BER gene network when *Mutyh* function is also impaired (or *Mutyh* expression is decreased). It is also possible that *D* alleles at *Ogg1* lead to a very subtle increase in C>A mutation rates, and we are simply underpowered to detect such a small mutation rate effect in the BXDs.

## No indication of causal structural variation or mobile element insertions near the chromosome 6 mutator locus

Although we argue above that *Ogg1* is likely the the best candidate gene to explain the new BXD C>A mutator phenotype, we cannot conclusively determine that the p.Thr95Ala missense mutation is a causal allele. We previously hypothesized that *Mutyh* missense mutations on *D* haplotypes were responsible for the large-effect C>A mutator phenotype we observed in the BXDs (*Sasani et al., 2022*). However, subsequent long-read assemblies of several inbred laboratory mouse strains revealed that this mutator phenotype might be caused by an ~5 kbp mobile element insertion (MEI) within the first intron of *Mutyh* (*Ferraj et al., 2023*), which is associated with significantly reduced expression of *Mutyh* in embryonic stem cells. We queried the new high-quality assemblies for evidence of mobile elements or other large SVs in the region surrounding the mutator locus on chromosome 6, but found no similarly compelling evidence that either SVs or MEIs might underlie the mutator phenotype described in this study.

## eQTLs might mediate germline mutator phenotypes in the BXDs

We observed strong-effect cis-eQTLs for *Ogg1* expression across a number of tissues in the BXDs (*Supplementary file 1*). In each of these tissue types, *D* genotypes were associated with decreased expression of *Ogg1*. As mentioned above, new evidence from long-read genome assemblies has demonstrated that an intronic MEI in *Mutyh* may be responsible for decreased *Mutyh* expression, and therefore higher C>A mutation rates, in BXDs with *D* haplotypes at the chromosome 4 mutator locus (*Ferraj et al., 2023*). Taken together, these results raise the exciting possibility that the mutator loci on both chromosome 4 and chromosome 6 lead to increased C>A mutation rates by lowering the expression of DNA repair genes in the same BER network. Moreover, the effect of the mutator locus on chromosome 6 may be conditional on the expression of the mutator near *Mutyh*, suggesting that complex genetic interactions can underlie germline mutator phenotypes in mammalian genomes.

### *Mbd4* may buffer the effects of *Mutyh* mutator alleles by triggering apoptosis

As mentioned in the *Results*, *Mbd4* lies just outside of the 90% bootstrap confidence interval on chromosome 6, but within a 10 Mbp window surrounding the AMSD peak. Due to the uncertainties associated with bootstrap confidence intervals in QTL mapping (*Manichaikul et al., 2006*), we have included a discussion of the evidence supporting *Mbd4* as a causal gene below.

Unlike the *Ogg1* p.Thr95Ala mutation, the p.Asp129Asn variant in *Mbd4* resides within an annotated protein domain (the *Mbd4* methyl-CpG binding domain), occurs at a nucleotide and amino acid residue that are both well conserved, and is predicted to be deleterious by SIFT (*Ng and Henikoff, 2003*, *Table 1*). A missense mutation that affects the homologous amino acid in humans (p.Asp142Gly in GRCh38/hg38) is also present on a single haplotype in the Genome Aggregation Database (gnomAD) (*Karczewski et al., 2020*) and is predicted by SIFT and Polyphen (*Adzhubei et al., 2010*) to be 'deleterious' and 'probably_damaging' in human genomes, respectively.

One puzzling observation is that loss-of-function mutations in *Mbd4* are not typically associated with C>A mutator phenotypes. Instead, *Mbd4* deficiency is usually implicated in C>T mutagenesis at CpG sites, and we did not detect an excess of C>T mutations in BXDs with *D* alleles at the chromosome 6 mutator locus (*Figure 3—figure supplement 1*). However, loss-of-function mutations in *Mbd4* have also been shown to exacerbate the effects of exogenous DNA damage agents. For example, mouse embryonic fibroblasts that harbor homozygous *Mbd4* KOs fail to undergo apoptosis following treatment with a number of chemotherapeutics and mutagenic compounds (*Cortellino et al., 2003*).

Most of these exogenous mutagens cause DNA damage that is normally repaired by mismatch repair (MMR) machinery, but murine intestinal cells with biallelic *Mbd4* LOF mutations also showed a reduced apoptotic response to gamma irradiation, which is repaired independently of the MMR gene *Mlh1* (*Sansom et al., 2003*). Homozygous loss of *Mbd4* function also leads to accelerated intestinal tumor formation in mice that harbor an *Apc* allele that predisposes them to intestinal neoplasia (*Millar et al., 2002*), and mice with biallelic truncations of the *Mbd4* coding sequence exhibit modestly increased mutation rates in colon cancer cell lines, including increased C>A mutation rates in certain lines (*Bader et al., 2007*).

Together, these lines of evidence suggest that *Mbd4* can modulate sensitivity to many types of exogenous mutagens, potentially through its role in determining whether cells harboring DNA damage should undergo apoptosis (*Cortellino et al., 2003*; *Sansom et al., 2003*). We speculate that in mice with deficient 8-oxoG repair – caused by a mutator allele in *Mutyh*, for example – ROS could cause accumulation of DNA damage in the germline. If those germ cells harbor fully functional copies of *Mbd4*, they might be able to trigger apoptosis and partially mitigate the effects of a *Mutyh* mutator allele. However, mice with reduced activity of both *Mbd4* and *Mutyh* may have a reduced ability to initiate cell death in response to DNA damage; as a result, their germ cells may accumulate even higher levels of ROS-mediated damage, leading to substantially elevated germline C>A mutation rates.

We anticipate that future experimental work will be needed to more conclusively establish a mechanistic explanation for the epistatic interaction between mutator loci described in this paper.

## Strengths and limitations of the AMSD approach

Our AMSD approach was able to identify a mutator allele that escaped notice using QTL mapping. To more systematically compare the power of AMSD and QTL mapping, we performed simulations under a variety of possible parameter regimes. Overall, we found that AMSD and QTL mapping have similar power to detect mutator alleles on haplotypes that each harbor tens or hundreds of de novo germline mutations (*Figure 1—figure supplement 2*). Nonetheless, only AMSD was able to discover the mutator locus on chromosome 6 in the BXDs, demonstrating that it outperforms QTL mapping in certain experimental systems. For example, simulations demonstrate that AMSD enjoys greater power than QTL mapping when haplotypes carry highly variable numbers of mutations that can be leveraged for mutator mapping (*Figure 1—figure supplement 3*). Because the BXDs were generated in six breeding epochs over a period of nearly 40 years, the oldest lines have accumulated orders of magnitude more mutations than the youngest lines; these younger BXDs have much noisier mutation spectra as a result. While approaches for QTL mapping typically weight the phenotypic measurements of each sample equally, AMSD compares the *aggregate* mutation spectra of haplotypes at every locus, a property that likely increased its power to detect mutators in the BXD dataset.

Another benefit of the AMSD approach is that it obviates the need to perform separate association tests for every possible $k$-mer mutation type, and therefore the need to adjust significance thresholds for multiple tests. Since AMSD compares the complete mutation spectrum between haplotypes that carry either allele at a site, it would also be well powered to detect a mutator allele that exerted a coordinated effect on multiple $k$-mer mutation types (e.g. increased the rates of both C>T and C>A mutations).

However, the AMSD method suffers a handful of drawbacks when compared to QTL mapping. Popular QTL mapping methods (such as R/qtl2; *Broman et al., 2019*) use linear models to test associations between genotypes and phenotypes, enabling the inclusion of additive and interactive covariates, as well as kinship matrices, in QTL scans. Although we have developed methods to account for inter-sample relatedness in the AMSD approach (Materials and methods), they are not as flexible as similar methods in QTL mapping software. Additionally, the AMSD method assumes that mutator alleles affect a subset of $k$-mer mutation types; if a mutator allele increased the rates of all mutation types equally on haplotypes that carried it, AMSD would be unable to detect it.

## Discovering mutator alleles in other experimental systems

Our discovery of a second BXD mutator allele underscores the power of RILs as a resource for dissecting the genetic architecture of germline mutation rates. Large populations of RILs exist for many model organisms, and we anticipate that as whole-genome sequencing becomes cheaper and cheaper, the

AMSD method could be useful for future mutator allele discovery outside of the BXDs. At the same time, RILs are a finite resource that require enormous investments of time and labor to construct. If germline mutator alleles are only detectable in these highly unusual experimental populations, we are unlikely to discover more than a small fraction of the mutator alleles that may exist in nature.

In natural, outbreeding populations, selection on germline mutator alleles will likely prevent large-effect mutators from reaching high allele frequencies, but a subset may be detectable by sequencing a sufficient number of human trios (*Milligan et al., 2022*). Since germline mutators often seem to exert their effects on a small number of mutation types, mutation spectrum analyses may have greater power to detect the genes that underlie heritable mutation rate variation, even if each gene has only a modest effect on the overall mutation rate per generation.

Thousands of human pedigrees have been sequenced in an effort to precisely estimate the rate of human de novo germline mutation (*Jónsson et al., 2017*; *Sasani et al., 2019*; *Werling et al., 2018*), and as family sequencing has become a more common step in the diagnosis of many congenital disorders, these datasets are growing on a daily basis. AMSD could potentially be applied to these large cohorts of two- or three-generation families. We propose that by pooling sparse mutation counts across many individuals who share the same candidate mutator allele, even a subtle mutator signal might rise above the noise of de novo germline mutation rate estimates. We note that the AMSD approach will require modification before it can be successfully applied to cohorts of outbred, sexually reproducing individuals. AMSD assumes that individuals harbor one of two possible genotypes at each marker, and does not yet account for heterozygous genotypes. As a result, our method is currently applicable only to resources like the BXD RILs, in which individuals have been inbred for sufficiently long that effectively all genotypes are homozygous. However, as human trio datasets become larger and larger in the coming years, rare mutator alleles may eventually be represented in as many individuals as the high-frequency variants present in the BXDs, thereby enabling more systematic mapping of elusive mutator phenotypes.

## Materials and methods
### Identifying de novo germline mutations in the BXDs

The BXD resource currently comprises a total of 152 RILs. BXDs were derived from either F2 or advanced intercrosses, and subsequently inbred by brother-sister mating for up to 180 generations (*Ashbrook et al., 2021*). BXDs were generated in distinct breeding 'epochs', which were each initiated with a distinct cross of C57BL/6J and DBA/2J parents; epochs 1, 2, 4, and 6 were derived from F2 crosses, while epochs 3 and 5 were derived from advanced intercrosses (*Ashbrook et al., 2021*). Previously, we analyzed whole-genome sequencing data from the BXDs and identified candidate de novo germline mutations in each line (*Sasani et al., 2022*). A detailed description of the methods used for DNA extraction, sequencing, alignment, and variant processing, as well as the characteristics of the de novo mutations, are available in a previous manuscript (*Sasani et al., 2022*).

Briefly, we identified private single-nucleotide mutations in each BXD that were absent from all other BXDs, as well as from the C57BL/6J and DBA/2J parents. We required each private variant to be meet the following criteria:

- genotyped as either homozygous or heterozygous for the alternate allele, with at least 90% of sequencing reads supporting the alternate allele;
- supported by at least 10 sequencing reads;
- Phred-scaled genotype quality of at least 20;
- must not overlap regions of the genome annotated as segmental duplications or simple repeats in GRCm38/mm10;
- must occur on a parental haplotype that was inherited by at least one other BXD at the same locus; these other BXDs must be homozygous for the reference allele at the variant site.

### A new approach to discover germline mutator alleles
#### Calculating AMSD

We developed a new approach to discover loci that affect the germline de novo mutation spectrum in biparental RILs (*Figure 1*).

We assume that a collection of haplotypes has been genotyped at informative markers, and that de novo germline mutations have been identified on each haplotype.

At each informative marker, we divide haplotypes into two groups based on the parental allele that they inherited. We then compute a $k$-mer mutation spectrum using the aggregate mutation counts in each haplotype group. The $k$-mer mutation spectrum contains the frequency of every possible $k$-mer mutation type in a collection of mutations, and can be represented as a vector of size $6 \times 4^{k-1}$ after collapsing by strand complement. For example, the 1-mer mutation spectrum is a six-element vector that contains the frequencies of C>T, C>G, C>A, A>G, A>T, and A>C mutations. Since C>T transitions at CpG nucleotides are often caused by a distinct mechanism (spontaneous deamination of methylated cytosine), we expand the 1-mer mutation spectrum to include a separate category for CpG>TpG mutations (*Arnheim and Calabrese, 2009*).

At each marker, we then compute the cosine distance between the two aggregate spectra. The cosine distance between two vectors $A$ and $B$ is defined as

$$D^C = 1 - \frac{A \cdot B}{\parallel A \parallel \parallel B \parallel}$$

where $\parallel A \parallel$ and $\parallel B \parallel$ are the $L^2$ (or Euclidean) norms of $A$ and $B$, respectively. The cosine distance metric has a number of favorable properties for comparing mutation spectra. Since it adjusts for the magnitude of the two input vectors, cosine distance can be used to compare two spectra with unequal total mutation counts (even if those total counts are relatively small). Additionally, by calculating the cosine distance between mutation spectra, we avoid the need to perform separate comparisons of mutation counts at each individual $k$-mer mutation type.

Inspired by methods from QTL mapping (*Broman et al., 2019*; *Churchill and Doerge, 1994*), we use permutation tests to establish genome-wide cosine distance thresholds. In each of $N$ permutation trials, we randomly shuffle the per-haplotype mutation data such that haplotype labels no longer correspond to the correct mutation counts. Using the shuffled mutation data, we perform a genome-wide scan as described above, and record the maximum cosine distance observed at any locus. After $N$ permutations (usually 10,000), we compute the $1 - p$ percentile of the distribution of maximum statistics, and use that percentile value as a genome-wide significance threshold (e.g. at $p = 0.05$).

## Estimating confidence intervals around AMSD peaks

If we identified an adjusted cosine distance peak on a particular chromosome, we used a bootstrap resampling approach (*Visscher et al., 1996*) to estimate confidence intervals. In each of $N = 10,000$ trials, we resampled the mutation spectrum data and corresponding marker genotypes (on the chromosome of interest) with replacement. Using those resampled spectra and genotypes, we performed an AMSD scan on the chromosome of interest and recorded the position of the marker with the largest adjusted cosine distance value. We then defined a 90% confidence interval by finding two marker locations between which 90% of all $N$ bootstrap samples produced a peak cosine distance value. In other words, we estimated the bounds of the 90% confidence interval by finding the markers that defined the 5th and 95th percentiles of the distribution of maximum adjusted cosine distance values across $N$ bootstrap trials. We note, however, that the bootstrap can exhibit poor performance in QTL mapping studies (*Manichaikul et al., 2006*); namely, bootstrap confidence intervals tend to be larger than those estimated using either an 'LOD drop' method or a Bayes credible interval, and can exhibit poorer-than-expected coverage (a measure of whether the confidence interval contains the true QTL location).

## Accounting for relatedness between strains

We expect each BXD to derive approximately 50% of its genome from C57BL/6J and 50% from DBA/2J. As a result, every pair of BXDs will likely have identical genotypes at a fraction of markers. Pairs of more genetically similar BXDs may also have more similar mutation spectra, potentially due to shared polygenic effects on the mutation process. Therefore, at a given marker, if the BXDs that inherited $D$ alleles are more genetically dissimilar from those that inherited $B$ alleles (considering all loci throughout the genome in our measurement of genetic similarity), we might expect the aggregate mutation spectra in the two groups to also be more dissimilar.

We implemented a simple approach to account for these potential issues of relatedness. At each marker $g_i$, we divide BXD haplotypes into two groups based on the parental allele they inherited. As before, we first compute the aggregate mutation spectrum in each group of haplotypes and calculate the cosine distance between the two aggregate spectra ($D_i^C$). Then, within each group of haplotypes, we calculate the allele frequency of the $D$ allele at every marker along the genome to obtain a vector of length $n$, where $n$ is the number of genotyped markers. To quantify the genetic similarity between the two groups of haplotypes, we calculate the Pearson correlation coefficient $r_i$ between the two vectors of marker-wide $D$ allele frequencies.

Put another way, at every marker $g_i$ along the genome, we divide BXD haplotypes into two groups and compute two metrics: $D_i^C$ (the cosine distance between the two groups' aggregate spectra) and $r_i$ (the correlation between genome-wide $D$ allele frequencies in the two groups). To control for the potential effects of genetic similarity on cosine distances, we regress $\left(D_1^C, D_2^C, \ldots D_n^C\right)$ on $(r_1, r_2, \ldots r_n)$ for all $n$ markers using an ordinary least-squares model. We then use the residuals from the fitted model as the 'adjusted' cosine distance values for each marker. If genome-wide genetic similarity between haplotypes perfectly predicts cosine distances at each marker, these residuals will all be 0 (or very close to 0). If genome-wide genetic similarity has no predictive power, the residuals will simply represent the difference between the observed cosine distance at a single marker and the marker-wide mean of cosine distances.

## Accounting for BXD population structure due to breeding epochs

The current BXD family was generated in six breeding 'epochs'. As discussed previously, each epoch was initiated with a distinct cross of C57BL/6J and DBA/2J parents; BXDs in four of the epochs were generated following F2 crosses of C57BL/6J and DBA/2J, and BXDs in the other two were generated following advanced intercrosses. Due to this breeding approach the BXD epochs differ from each other in a few important ways. For example, BXDs derived in epochs 3 and 5 (i.e. from advanced intercross) harbor larger numbers of fixed recombination breakpoints than those from epochs 1, 2, 4, and 6 (*Ashbrook et al., 2021*). Although the C57BL/6J and DBA/2J parents used to initialize each epoch were completely inbred, they each possessed a small number unique de novo germline mutations that were subsequently inherited by many of their offspring. A number of these 'epoch-specific' variants have also been linked to phenotypic variation observed between BXDs from different epochs (*Ashbrook et al., 2021*; *Stafford et al., 2019*; *Shoucri et al., 2017*; *Lu et al., 2019*).

To account for potential population structure, as well as these epoch-specific effects, we introduced the ability to perform stratified permutation tests in the AMSD approach. Normally, in each of $N$ permutations we shuffle the per-haplotype mutation spectrum data such that haplotype labels no longer correspond to the correct mutation spectra (i.e. shuffle mutation spectra *across* epochs). In the stratified approach, we instead shuffle per-haplotype mutation data *within* epochs, preserving epoch structure while still enabling mutation spectra permutations. We used this epoch-aware approach for all permutation tests presented in this manuscript.

## Implementation and source code

The AMSD method was implemented in Python, and relies heavily on the following Python libraries: `numpy`, `pandas`, `matplotlib`, `scikit-learn`, `pandera`, `seaborn`, and `numba` (*Harris et al., 2020*; *The Pandas Development Team, 2023*; *Hunter, 2007*; *Pedregosa et al., 2011*; *Bantilan, 2020*; *Waskom, 2021*; *Finkel, 2015*).

The code underlying AMSD, as well as documentation of the method, is available on GitHub under the MIT license, and is archived at Zenodo (*Sasani, 2024*). We have also deposited a reproducible Snakemake workflow (*Kosmidis et al., 2020*) for running reproducing all analyses and figures presented in the manuscript.

## **Simulations to assess the power of the AMSD approach**

We performed a series of simple simulations to estimate our power to detect alleles that affect the germline mutation spectrum using the AMSD method.

## Simulating genotypes

First, we simulate genotypes on a population of haplotypes at a collection of sites. We define a matrix $G$ of size $(s, h)$, where $s$ is the number of sites and $h$ is the number of haplotypes. We assume that every site is biallelic, and that the minor allele frequency at every site is 0.5. For every entry $G_{i,j}$, we take a single draw from a uniform distribution in the interval $[0.0, 1.0]$. If the value of that draw is less than 0.5, we assign the value of $G_{i,j}$ to be 1. Otherwise, we assign the value of $G_{i,j}$ to be 0.

## Defining expected mutation type probabilities

Next, we define a vector of 1-mer mutation probabilities:

$$P = (0.29, 0.17, 0.12, 0.075, 0.1, 0.075, 0.17)$$

These probabilities sum to 1 and roughly correspond to the expected frequencies of C>T, CpG>TpG, C>A, C>G, A>T, A>C, and A>G de novo germline mutations in mice, respectively (**Lindsay et al., 2019**). If we are simulating the 3-mer mutation spectrum, we modify the vector of mutation probabilities $P$ to be length 96, and assign every 3-mer mutation type a value of $\frac{P_c}{16}$, where $P_c$ is the probability of the 'central' mutation type associated with the 3-mer mutation type. In other words, each of the 16 possible NCN>NTN 3-mer mutation types would be assigned a mutation probability of $\frac{P_c}{16} = \frac{0.46}{16} = 0.02875$. We then generate a vector of lambda values by scaling the mutation probabilities by the number of mutations we wish to simulate ($m$):

$$\lambda = Pm$$

We also create a second vector of lambda values ($\lambda'$), in which we multiply the $\lambda$ value of a single mutation type by the mutator effect size $e$.

Rather than simulating the same mean number of mutations ($m$) on every haplotype, we also performed a series of simulations in which the mean number of mutations on each haplotype was allowed to vary. The BXD RILs were inbred for variable numbers of generations, and each BXD therefore accumulated a variable number of de novo germline mutations (**Sasani et al., 2022**). To more closely approximate the BXD haplotypes, we performed simulations in which the number of mutations ($m$) on each haplotype was drawn from a uniform distribution from $m$ to $20m$. In other words, we created a vector of mutation counts $M$ containing $h$ evenly spaced integers from $m$ to $20m$, where $h$ is the number of simulated haplotypes. Thus, if we simulated between 100 and 2000 mutations on 50 haplotypes, the $i$th entry of $M$ would be $100 + \frac{(2000-100)}{50}i$. Each haplotype's mean number of mutations was then assigned by looking up the haplotype's index $i$ in $M$.

In our simulations, we assume that genotypes at a single site (the 'mutator locus') are associated with variation in the mutation spectrum. That is, at a single site $s_i$, all of the haplotypes with 1 alleles should have elevated rates of a particular mutation type and draw their mutation counts from $\lambda'$, while all of the haplotypes with 0 alleles should have 'wild-type' rates of that mutation type and draw their mutation counts from $\lambda$. We therefore pick a random site $s_i$ to be the 'mutator locus', and identify the indices of haplotypes in $G$ that were assigned 1 alleles at $s_i$. We call these indices $h_{mut}$.

## Simulating mutation spectra

To simulate the mutation spectrum on our toy population of haplotypes, we define a matrix $C$ of size $(h, n)$, where $n = 6 \times 4^{k-1}$ (or if $k = 1$ and we include CpG>TpG mutations, $6 \times 4^{k-1} + 1$).

Then, we populate the matrix $C$ separately for *mutator* and *wild-type* haplotypes. For every row $i$ in the matrix (i.e. for every haplotype), we first ask if $i$ is in $h_{mut}$ (i.e. if the haplotype at index $i$ was assigned a 1 allele at the 'mutator locus'). If so, we set the values of $C_i$ to be the results of a single Poisson draw from $\lambda'$. If row $i$ is not in $h_{mut}$, we set the values of $C_i$ to be the results of a single Poisson draw from $\lambda$.

## Assessing power to detect a simulated mutator allele using AMSD

For each combination of parameters (number of simulated haplotypes, number of simulated markers, mutator effect size, etc.), we run 100 independent trials. In each trial, we simulate the genotype matrix $G$ and the mutation counts $C$. We calculate a 'focal' cosine distance as the cosine distance between the

aggregate mutation spectra of haplotypes with either genotype at $s_i$ (the site at which we artificially simulated an association between genotypes and mutation spectrum variation). We then perform an AMSD scan using $N$=1000 permutations. If fewer than 5% of the $N$ permutations produced a cosine distance greater than or equal to the focal distance, we say that the approach successfully identified the mutator allele in that trial.

## Assessing power to detect a simulated mutator allele using QTL mapping

Using simulated data, we also assessed the power of traditional QTL mapping to detect a locus associated with mutation spectrum variation. As described above, we simulated both genotype and mutation spectra for a population of haplotypes under various conditions (number of mutations per haplotype, mutator effect size, etc.). Using those simulated data, we used R/qtl2 (*Broman et al., 2019*) to perform a genome scan for significant QTL as follows: we assume that the simulated genotype markers are evenly spaced (in physical Mbp coordinates) on a single chromosome. First, we calculate the fraction of each haplotype's de novo mutations that belong to each of the $6 \times 4^{k-1}$ possible $k$-mer mutation types. We then convert the simulated genotypes at each marker to genotype probabilities using the `calc_genoprob` function in R/qtl2, with `map_function = "c-f"` and `error_prob =0`. For every $k$-mer mutation type, we use genotype probabilities and per-haplotype mutation fractions to perform a scan for QTL with the scan1 function; to make the results more comparable to those from the AMSD method, we do not include any covariates or kinship matrices in these QTL scans. We then use the `scan1perm` function to perform 1000 permutations of the per-haplotype mutation fractions and calculate LOD thresholds for significance. We consider the QTL scan to be 'successful' if it produces an LOD score above the significance threshold (defined using $\alpha = \frac{0.05}{7}$) for the marker at which we simulated an association with mutation spectrum variation.

In our simulations, we augment the mutation rate of a single $k$-mer mutation type on haplotypes carrying the simulated mutator allele. However, in an experimental setting, we would not expect to have a priori knowledge of the mutation type affected by the mutator. Thus, by using an alpha threshold of 0.05 in our simulations, we would likely over-estimate the power of QTL mapping for detecting the mutator. Since we would need to perform seven separate QTL scans (one for each 1-mer mutation type plus CpG>TpG) in an experimental setting, we calculate QTL LOD thresholds at a Bonferroni-corrected alpha value of $\alpha = \frac{0.05}{7}$ .

## Applying the AMSD method to the BXDs

We downloaded previously generated BXD de novo germline mutation data from the GitHub repository associated with our previous manuscript, which was also archived at Zenodo (*Sasani et al., 2022*; *Sasani, 2022*), and downloaded a CSV file of BXD genotypes at ~7300 informative markers from GeneNetwork (*Mulligan et al., 2017*; *BXD Genotype, 2017*). We also downloaded relevant metadata about each BXD from the manuscript describing the updated BXD resource (*Ashbrook et al., 2021*). These files are included in the GitHub repository associated with this manuscript.

As in our previous manuscript (*Sasani et al., 2022*), we included mutation data from a subset of the 152 BXDs in our AMSD scans. Specifically, we removed BXDs that were backcrossed to a C57BL/6J or DBA/2J parent at any point during the inbreeding process (usually, in order to rescue that BXD from inbreeding depression; *Ashbrook et al., 2021*). We also removed BXD68 from our genome-wide scans, since we previously discovered a hyper-mutator phenotype in that line; the C>A germline mutation rate in BXD68 is over five times the population mean, likely due to a private deleterious nonsynonymous mutation in *Mutyh* (*Sasani et al., 2022*). In our previous manuscript, we removed any BXDs that had been inbred for fewer than 20 generations, as it takes approximately 20 generations of strict brother-sister mating for an RIL genome to become >98% homozygous (*Green, 1981*). As a result, any potential mutator allele would almost certainly be either fixed or lost after 20 generations; if fixed, the allele would remain linked to any excess mutations it causes for the duration of subsequent inbreeding. In other words, the de novo mutations present in the genome of a 'young' BXD (i.e. a BXD that was inbred for fewer than 20 generations) would not reflect a mutator allele's activity as strongly as the mutations present in the genome of a much older BXD. This presented a challenge when we used QTL mapping to discover mutator alleles in our previous manuscript, since the phenotypes (i.e. C>A mutation rates) of young and old BXDs were weighted equally; thus, we simply removed the younger BXDs from our analysis to avoid using their especially noisy mutation spectra. Since AMSD

computes an *aggregate* mutation spectrum using all BXDs that inherited a particular allele at a locus, and can overcome the sparsity and noise of individual mutation spectra, we chose to include these younger BXDs in our genome-wide scans in this study.

In total, we included 117 BXDs in our genome-wide scans.

### Identifying candidate single-nucleotide mutator alleles overlapping the chromosome 6 peak

We investigated the region implicated by our AMSD approach on chromosome 6 by subsetting the joint-genotyped BXD VCF file (European Nucleotide Archive accession PRJEB45429; *Sasani et al., 2021*) using bcftools (*Danecek et al., 2021*). We defined the candidate interval surrounding the cosine distance peak on chromosome 6 as the 90% bootstrap confidence interval (extending from approximately 95 to 114 Mbp). To predict the functional impacts of both single-nucleotide variants and indels on splicing, protein structure, etc., we annotated variants in the BXD VCF using the following `snpEff` (*Cingolani et al., 2012*) command:

```
java -Xmx16g -jar /path/to/snpeff/jarfile GRCm38.75/path/to/bxd/vcf > /path/
to/uncompressed/output/vcf
```

and used `cyvcf2` (*Pedersen and Quinlan, 2017*) to iterate over the annotated VCF file in order to identify nonsynonymous fixed differences between the parental C57BL/6J and DBA/2J strains.

### Identifying candidate SV alleles overlapping the chromosome 6 peak

We downloaded summary VCFs containing insertion, deletion, and inversion SVs (identified via high-quality, long-read assembly of inbred laboratory mouse strains; *Ferraj et al., 2023*) from the Zenodo link associated with the Ferraj et al. manuscript: https://doi.org/10.5281/zenodo.7644286.

We then downloaded a TSV file containing RefSeq gene predictions in GRCm39/mm39 from the UCSC Table Browser (*Karolchik et al., 2004*), and used the `bx-python` library (*Taylor lab, 2023*) to intersect the interval spanned by each SV with the intervals spanned by the `txStart` and `txEnd` of every RefSeq entry. We queried all SVs within the 90% bootstrap confidence interval on chromosome 6.

### Extracting mutation signatures

We used SigProfilerExtractor (v.1.1.21) (*Tate et al., 2019*) to extract mutation signatures from the BXD mutation data. After converting the BXD mutation data to the 'matrix' input format expected by SigProfilerExtractor, we ran the `sigProfilerExtractor` method as follows:

```
# install the mm10 mouse reference data
genInstall.install('mm10')

# run mutation signature extraction
sig.sigProfilerExtractor(
  'matrix',
  /path/to/output/directory,
  /path/to/input/mutations,
  maximum_signatures=10,
  nmf_replicates=100,
  opportunity_genome="mm10",
)
```

### Comparing mutation spectra between MGP strains

We downloaded mutation data from a previously published analysis (*Dumont, 2019*, their Supplementary file 1, Excel Table S3) that identified strain-private mutations in 29 strains that were originally whole-genome sequenced as part of the Sanger MGP (*Keane et al., 2011*). When comparing counts of each mutation type between MGP strains that harbored either *D* or *B* alleles at the chromosome 4

**Table 2.** Names of gene expression datasets used for each tissue type on GeneNetwork.

| Tissue name | Complete name of GeneNetwork expression data |
|---|---|
| Kidney | `Mouse kidney M430v2 Sex Balanced (Aug06) RMA` |
| Gastrointestinal | `UTHSC Mouse BXD Gastrointestinal Affy MoGene 1.0 ST Gene Level (Apr14) RMA` |
| Hematopoetic stem cells | `UMCG Stem Cells ILM6v1.1 (Apr09) transformed` |
| Spleen | `UTHSC Affy MoGene 1.0 ST Spleen (Dec10) RMA` |
| Liver | `UTHSC BXD Liver RNA-Seq Avg (Oct19) TPM Log2` |
| Heart | `NHLBI BXD All Ages Heart RNA-Seq (Nov20) TMP Log2 **` |
| Hippocampus | `Hippocampus Consortium M430v2 (Jun06) RMA` |

or chromosome 6 mutator loci, we adjusted mutation counts by the number of callable A, T, C, or G nucleotides in each strain as described previously (*Sasani et al., 2022*).

## Querying GeneNetwork for eQTLs at the mutator locus

We used the online GeneNetwork resource (*Mulligan et al., 2017*), which contains array- and RNA-seq-derived expression measurements in a wide variety of tissues, to find *cis*-eQTLs for the DNA repair genes we implicated under the cosine distance peak on chromosome 6. On the GeneNetwork homepage (genenetwork.org), we selected the 'BXD Family' Group and used the Type dropdown menu to select each of the specific expression datasets described in *Table 2*. In the Get Any text box, we then entered the listed gene name and clicked Search. After selecting the appropriate trait ID on the next page, we used the Mapping Tools dropdown to run Hayley-Knott regression (*Haley and Knott, 1992*) with default parameters: 1000 permutations, interval mapping, no cofactors, and WGS-based genotypes (2022).

If we discovered a significant cis-eQTL for the gene of interest (i.e. a locus on chromosome 6 with an LRS greater than or equal to the 'significant LRS' genome-wide threshold), we then performed a second genome-wide association test for the trait of interest using GEMMA (*Girolami et al., 1987*) with the following parameters: WGS-based marker genotypes, a minor allele frequency threshold of 0.05, and leave-one-chromosome-out. By using both Haley-Knott regression and GEMMA, we could first discover loci that exceeded a genome-wide LRS threshold, and then more precisely estimate the effect of those loci on gene expression (*Watson and Ashbrook, 2020*).

The exact names of the expression datasets we used for each tissue are shown in *Table 2*.

## Calculating the frequencies of candidate mutator alleles in wild mice

To determine the frequencies of the *Ogg1* and *Setmar* nonsynonymous mutations in other populations of mice, we queried a VCF file containing genome-wide variation in 67 wild-derived mice from four species of *Mus* (*Harr et al., 2016*). We calculated the allele frequency of each nonsynonymous mutation in each of the four species or subspecies (*M. musculus domesticus*, *M. musculus musculus*, *M. musculus castaneus*, and *M. spretus*), including genotypes that met the following criteria:

- supported by at least 10 sequencing reads;
- Phred-scaled genotype quality of at least 20.

## Testing for epistasis between the two mutator loci

To test for statistical epistasis between the mutator loci on chromosome 4 and chromosome 6, we modeled C>A mutation rates in the BXDs as a function of genotypes at either locus:

$$\log\lambda^i = \beta_0 + \beta_1 X_1^i + \beta_2 X_2^i + \log A^i$$

Here, $\lambda^i$ represents the count of C>A mutations in the $i$th BXD. We assume that the count of C>A mutations follows a Poisson distribution with a mean equal to the C>A mutation rate (expressed per base pair, per generation) multiplied by product of two terms: the total number of generations of inbreeding and the total number of base pairs accessible to variant calling. The BXDs differ in both their durations of inbreeding and the proportions of their genomes that were sequenced to sufficient depth, which influences the number of mutations we observe in each BXD. We therefore model C>A mutation counts as *rates* by including a $\log A^i$ term – sometimes referred to as an 'offset' – which represents the product of the number of 'callable' cytosine/guanine nucleotides in each BXD (i.e. the total number of cytosines/guanines covered by at least 10 sequencing reads) and the number of generations for which the BXD was inbred. The offset is assumed to have a coefficient of 1. The $X_1^i$ and $X_2^i$ terms represent BXD genotypes in the $i$th BXD at markers rs27509845 and rs46276051 (the markers with peak cosine distances on chromosomes 4 and 6 in the two AMSD scans); *B* genotypes are coded as 0 and *D* genotypes are coded as 1. We limited our analysis to the $n$ = 108 BXDs that were homozygous at both loci, allowing us to model genotypes at either locus as binary categorical variables. In the R statistical language, we defined the model as follows:

```
m1 <-glm(Count ~offset(log(ADJ_AGE))+Genotype_
A+Genotype_B,,data=data,,family=poisson())
```

In this model, `Count` is the count of C>A de novo mutations observed in each BXD. The `Genotype_A` and `Genotype_B` terms are analogous to the *X*1 and *X*2 terms in the mathematical model above, and ADJ_AGE is analogous to the *A* term.

To explicitly test if the effects of *D* genotypes at the chromosome 6 locus depended on the presence of *D* genotypes at the chromosome 4 locus, we fit a second model incorporating a multiplicative interaction between $X_1$ and $X_2$:

$$\log\lambda^i = \beta_0 + \beta_1 X_1^i + \beta_2 X_2^i + \beta_{12} X_1^i X_2^i + \log A^i$$

Here, the $\beta_{12}$ term captures the interaction effect between genotypes at the two mutator loci. If both $X_1^i$ and $X_2^i$ are equal to 1 (i.e. genotypes at both the chromosome 4 and chromosome 6 loci are *D*), the $\beta_{12}$ term will capture the resulting effect on C>A mutation counts. We can interpret a significantly non-zero $\beta_{12}$ value as evidence for a non-additive effect of *D* genotypes at both loci.

We defined this model in the R statistical language as follows:

```
m2 <-glm(Count ~offset(log(ADJ_AGE))+Genotype_A *
Genotype_B,,data=data,,family=poisson())
```

An identical model can also be written by explicitly specifying both the additive and interaction effects:

```
m2 <-glm(Count ~offset(log(ADJ_AGE))+Genotype_A+Genotype_
B+Genotype_A:Genotype_B, data=data,,family=poisson())
```

Using analysis of variance (ANOVA), we compared the model incorporating an interaction effect to the one with only additive effects:

```
anova(m2, m1, test="Chisq")
```

If model m2 is a significantly better fit to the data than m1, we can reject the null hypothesis that the effect of *D* genotypes at both markers is equal to the sum of the marginal effects of *D* genotypes at either `rs27509845` or `rs46276051`. In other words, if m2 is a better fit than m1, then the combined effect of *D* genotypes at both markers is non-additive, and indicative of statistical epistasis.

We tested for statistical epistasis in the Sanger MGP strains using a nearly identical approach. In this analysis, we fit two models as follows:

```
m1 <-glm(Count ~offset(log(CALLABLE_C))+Genotype_A *
Genotype_B,,data=data,,family=poisson())
```

```
m2 <-glm(Count ~offset(log(CALLABLE_C))+Genotype_
A+Genotype_B,,data=data,,family=poisson())
```

where Count is the count of strain-private C>A mutations observed in each MGP strain (*Dumont, 2019*). The `CALLABLE_C` term represents the total number of cytosine and guanine nucleotides that were accessible for mutation calling in each strain, and the Genotype_A and Genotype_B terms represent MGP genotypes at the chromosome 4 and chromosome 6 mutator loci, respectively. We compared the two models using ANOVA as described above.

## Acknowledgements

We thank Robert W Williams (University of Tennessee Health Sciences Center) and Don F Conrad (Oregon Health & Science University) for very helpful comments and feedback on a draft of this manuscript.

## Additional information

### Funding

| Funder | Grant reference number | Author |
| --- | --- | --- |
| National Institute of General Medical Sciences | R35GM133428 | Kelley Harris |
| Burroughs Wellcome Fund | Career Award at the Scientific Interface | Kelley Harris |
| Kinship Foundation | Searle Scholarship | Kelley Harris |
| Pew Charitable Trusts | Pew Biomedical Scholarship | Kelley Harris |
| Alfred P. Sloan Foundation | Sloan Fellowship | Kelley Harris |
| Allen Discovery Center | Discovery Center for Cell Lineage Tracing | Kelley Harris |
| National Human Genome Research Institute | R01HG012252 | Aaron R Quinlan |

The funders had no role in study design, data collection and interpretation, or the decision to submit the work for publication. For the purpose of Open Access, the authors have applied a CC BY public copyright license to any Author Accepted Manuscript version arising from this submission.

### Author contributions

Thomas A Sasani, Conceptualization, Data curation, Software, Formal analysis, Investigation, Visualization, Methodology, Writing – original draft, Writing – review and editing; Aaron R Quinlan, Conceptualization, Resources, Supervision, Funding acquisition, Methodology, Writing – review and editing; Kelley Harris, Conceptualization, Resources, Supervision, Funding acquisition, Methodology, Writing – original draft, Writing – review and editing

### Author ORCIDs

Thomas A Sasani ⓘ https://orcid.org/0000-0003-2317-1374
Kelley Harris ⓘ http://orcid.org/0000-0003-0302-2523

Reviewer #1 (Public Review): https://doi.org/10.7554/eLife.89096.3.sa1
Reviewer #2 (Public Review): https://doi.org/10.7554/eLife.89096.3.sa2
Reviewer #3 (Public Review): https://doi.org/10.7554/eLife.89096.3.sa3
Author Response https://doi.org/10.7554/eLife.89096.3.sa4

## Additional files

### Supplementary files

• Supplementary file 1. Significant cis-eQTLs (expression quantitative trait loci) for DNA repair genes in various tissues identified using GeneNetwork.

• MDAR checklist

### Data availability

The current manuscript is a computational study, so no new data have been generated. All source code is available on GitHub and is archived on Zenodo. The code underlying AMSD, as well as documentation of the method, is available on GitHub under the MIT license, and is archived at Zenodo (*Sasani, 2024*). We have also deposited a reproducible Snakemake workflow for running reproducing all analyses and figures presented in the manuscript. BXD germline de novo mutation data generated in *Sasani et al., 2022* can be downloaded from GitHub and are additionally archived at Zenodo (*Sasani, 2022*). Raw BXD variant calls are available in VCF format from the European Nucleotide Archive (project accession PRJEB45429). Strain-private mutation data from the Sanger MGP samples are available as supplementary data from *Dumont, 2019*. Assembly-derived SV calls from inbred laboratory strains are available as supplementary data from *Ferraj et al., 2023*. BXD gene expression data were accessed from GeneNetwork.

The following previously published dataset was used:

| Author(s) | Year | Dataset title | Dataset URL | Database and Identifier |
|---|---|---|---|---|
| Sasani TA, Ashbrook DG, Beichman AC, Lu L, Palmer AA, Williams RW, Pritchard JK, Harris K | 2021 | Data from: A natural mutator allele shapes mutation spectrum variation in mice | https://www.ebi.ac.uk/ena/browser/view/PRJEB45429 | European Nucleotide Archive, PRJEB45429 |

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

## Appendix 1

### Missense mutations in *Setmar* are unlikely to contribute to epistasis in the BXDs

Unlike *Ogg1*, *Setmar* does not participate directly in BER, though its primate ortholog plays an indirect role in the repair of double-stranded DNA breaks via NHEJ. In anthropoid primates, *SETMAR* encodes a fusion of two functional domains: a SET domain-containing histone methyltransferase and a transposase domain from the *Mariner* family (MAR) (*Cordaux et al., 2006*); the mouse *Setmar* ortholog only encodes the histone methyltransferase domain. In human cell lines, *SETMAR* localizes to induced DSBs and dimethylates nearby H3K36, which promotes the recruitment of DNA repair components involved in NHEJ to the DSB (*Fnu et al., 2011*). There is also evidence that overexpression of *SETMAR* (also known as *Metnase*) improves the efficiency of NHEJ (*Lee et al., 2005*) and leads to increased cell survival following exposure to ionizing radiation (*Lee et al., 2005*). Point mutations in either the SET or MAR domains significantly reduced the ability of *SETMAR* to promote NHEJ and DNA repair (*Lee et al., 2005*; *Kim et al., 2014*; *Beck et al., 2011*), suggesting that both domains are needed for its role in DNA repair. Another study found that overexpression of the isolated SET and MAR domains, but not of wild-type *SETMAR*, had a modest effect on NHEJ repair; overexpression of the SET domain slightly *decreased* NHEJ repair of a linearized plasmid in human cells, while overexpression of the *Mariner*-derived domain increased NHEJ relative to controls (*Tellier and Chalmers, 2019*).

Taken together, these results suggest that both the SET and transposase domains of primate *SETMAR* are important for *SETMAR*-mediated DNA repair. The p.Leu103Phe missense mutation that differentiates C57BL/6J and DBA/2J (*Table 1*) resides within the *Setmar* pre-SET domain and occurs at an amino acid residue that is predicted to be deleterious by SIFT (*Adzhubei et al., 2010*). However, since the mouse *Setmar* ortholog lacks the *Mariner*-derived domain, we believe that the the p.Leu103Phe or p.Ser273Arg missense mutations are unlikely to affect C>A mutation rates in the BXDs. Moreover, we believe that the documented mutator phenotypes associated with *Ogg1*, as well as that gene's known role in BER, make it more likely candidate to underlie the epistatic interaction with *Mutyh* we observed in this study.

