## [Editor Report · eLife assessment]

By developing a novel method for detecting genetic variants associated with germline mutation spectrum variation, this **important** study identifies a new "mutator" locus in a population of inbred mouse strains, although the causal gene(s) and allele(s) within this locus remain uncertain. The authors further demonstrate that this new mutator locus interacts epistatically with a previously identified mutator allele on C>A mutation rate, showcasing the complexity of the genetic basis underlying variation in mutation rate and spectrum. Evidence for major findings in this paper is **convincing**, and the new method has the potential to be applicable to a variety of experimental systems and natural populations.

---

## [Referee Report · Reviewer #1 (Public Review)]

The mutation rate and spectrum have been found to differ between populations as well as across individuals within the same population. Hypothesizing that some of the observed variation has a genetic basis, the authors of this paper have made important contributions in the past few years in identifying genetic variants that modify mutation rate or spectrum in natural populations. This paper makes one significant step further by developing a new method for mapping genetic variants associated with the mutation spectrum, which reveals new biological insights.

Using traditional quantitative trait locus (QTL) mapping in the BXD mouse recombinant inbred lines (RILs), the authors of this paper previously identified a genetic locus associated with C>A mutation rate. However, this approach has limited power, as it suffers from multiple testing burden as well as noise in the "observed mutation spectrum phenotype" due to rarity and randomness of mutation events. To overcome these limitations, the authors developed a new method that they named "aggregate mutation spectrum distance" (AMSD), which in short measures the difference in the aggregate mutation spectrum between two groups of individuals with distinct genotypes at a specific genomic locus. With this new approach, they recover the previously reported candidate mutator locus (near Mutyh gene) and identify a new candidate locus that modifies the C>A mutation rate on only the mutator allele genetic background at the Mutyh locus. Using more rigorous statistical testing, the authors show convincingly synergistic epistatic effects between the mutator alleles at the two loci.

Overall, the analyses presented are well done and provide convincing evidence for the major findings, including the new candidate mutator locus and its epistatic interaction with the Mutyh locus. The new AMSD method introduced is innovative and outperforms traditional QTL mapping under most conditions, as demonstrated by extensive simulations. I identify no major issues with this paper and think it is very well written.

One of the major advantages of the AMSD method over QTL mapping is alleviation of the multiple testing burden, as one comparison tests for any changes in the mutation spectrum, including simultaneous, small changes in the relative abundance of multiple mutation types. The flip side of this advantage of AMSD is that, when a significant association is detected, it is not immediately clear which mutation type is driving the signal. To narrow the signal to specific candidate mutation type(s), additional analyses are needed, such as testing for differential proportions of each mutation type between individuals with or without the candidate mutator allele. However, such analysis may be less powerful when the mutator allele leads to small changes in the relative abundance of multiple mutation types. This will be an area of improvement for future studies.

---

## [Referee Report · Reviewer #2 (Public Review)]

In this paper Sasani, Quinlan and Harris present a new method for identifying genetic factors affecting germline mutation, which is particularly applicable to genome sequence data from mutation accumulation experiments using recombinant inbred lines. These are experiments where laboratory organisms are crossed and repeatedly inbred for many generations, to build up a substantial number of identifiable germline mutations. The authors apply their method to such data from mice, and identify two genetic factors at two separate genetic loci. Clear evidence of such factors has been difficult to obtain, so this is an important finding. They further show evidence of an epistatic interaction between these factors (meaning that they do not act independently in their effects on the germline mutation process). This is exciting because such interactions are difficult to detect and few if any other examples have been studied.

The authors present a careful comparison of their method to another similar approach, quantitative trait locus (QTL) analysis, and demonstrate that in situations such as the one analysed it has greater power to detect genetic factors with a certain magnitude of effect. They also test the statistical properties of their method using simulated data and permutation tests. Overall the analysis is rigorous and well motivated, and the methods explained clearly.

The main limitation of the approach is that it is difficult to see how it might be applied beyond the context of mutation accumulation experiments using recombinant inbred lines. This is because the signal it detects, and hence its power, is based on the number of extra accumulated mutations linked to (i.e. on the same chromosome as) the mutator allele. In germline mutation studies of wild populations the number of generations involved (and hence the total number of mutations) is typically small, or else the mutator allele becomes unlinked from the mutations it has caused (due to recombination), or is lost from the population altogther (due to chance or perhaps selection against its deleterious consequences).

Nevertheless, accumulation lines are a common and well established experimental approach to studying mutation processes in many organisms, so the new method could have wide application and impact on our understanding of this fundamental biological process.

The evidence presented for an epistatic interaction is convincing, and the authors suggest some plausible potential mechanisms for how this interaction might arise, involving the DNA repair machinery and based on previous studies of the proteins implicated. However as with all such findings, given the higher degree of complexity of the proposed model it needs to be treated with greater caution, perhaps until replicated in a separate dataset or demonstrated in follow-up experiments exploring the pathway itself.

---

## [Referee Report · Reviewer #3 (Public Review)]

Sasani et al. develop and implement a new method for mutator allele discovery in the BXD mouse population. This new method, termed "aggregate mutation spectrum distance" or AMSD, carries several notable strengths, including the ability to aggregate de novo mutations across individuals to reduce data sparsity and to combine mutation rate frequencies across multiple nucleotide contexts into a single estimate. As demonstrated by simulations, this method is better suited to mutator discovery under certain scenarios, as compared to conventional QTL or association mapping. Overall, the theoretical premise of the AMSD method is judged to be both strong and innovative, and the methodology could be extended to other species and populations to enable discovery of additional mutator alleles.

The authors then apply their method to the BXD mouse recombinant inbred mapping population. As proof-of-principle, they first successfully re-identify a known mutator locus in this population on chr4. Next, to assess possible genetic interactions involving this known mutator, Sasani et al. condition on the chr4 mutator genotype and reimplement the AMSD scan. This strategy led them to identify a second locus on chr6 that interacts epistatically with the chr4 locus; mice with "D" alleles at both loci exhibit a significantly increased burden of C>A de novo mutations, even though mice with the D allele at the chr6 locus alone show no appreciable increase in the C>A mutation fraction. This exciting discovery not only adds to the catalog of known mutator alleles, but also reveals key aspects of mutator biology and reinforces the hypothesis that segregating variants in genes associated with DNA repair influence germline mutation spectra.

Despite a high level of overall enthusiasm for this work, there are some limitations to the AMSD method. However, it is my judgement that the authors present a balanced summary of the strengths and weaknesses of their method in the revised manuscript. I also think that the authors' conclusions may actually somewhat undersell the scientific impact of their findings. As the authors note, few mutation rate modifiers have been identified in mammals. This is potentially because large- and moderate-effect modifiers are rapidly selected against due to their deleterious effects, but could also be due to pervasive epistasis wherein modifiers are only expressed on certain "permissive" genetic backgrounds, such as the chr6 locus the authors discover in this paper. The potential background dependence of mutator expression could partially shelter it from the action of selection, allowing the allele persist in populations. This discovery has significant implications for our understanding of mutation rate evolution, but only earns a cursory mention in the paper.

---

## [Author Response]

The following is the authors’ response to the original reviews.

**Reviewer 1 (Public Review):**
1. The name of the new method "inter-haplotype distance" is more confusing than helpful, as the haplotype information is not critical for implementing this method. First, the mutation spectrum is aggregated genome-wide regardless of the haplotypes where the mutations are found. Second, the only critical haplotype information is that at the focal site (i.e., the locus that is tested for association): individuals are aggregated together when they belong to the same "haplotype group" at the focal site. However, for the classification step, haplotype information is not really necessary: individuals can be grouped based on their genotypes at the given locus (e.g., AA vs AB). As the authors mentioned, this method can be potentially applied to other mutation datasets, where haplotype information may well be unavailable. I hope the authors can reconsider the name and remove the term "haplotype" (perhaps something like "inter-genotype distance"?) to avoid giving the wrong impression that haplotype information is critical for applying this method.

We appreciate the reviewer's concern about the name of our method. The reviewer is correct that haplotype information is not critical for our method to work, and as a result we've decided to simply rename the approach to "aggregate mutation spectrum distance" (abbreviated AMSD). For simplicity, we refer to the method as IHD throughout our responses to reviewers, but the revised manuscript now refers to AMSD.

1. The biggest advantage of the IHD method over QTL mapping is alleviation of the multiple testing burden, as one comparison tests for any changes in the mutation spectrum, including simultaneous, small changes in the relative abundance of multiple mutation types. Based on this, the authors claim that IHD is more powerful to detect a mutator allele that affects multiple mutation types. Although logically plausible, it is unclear under what quantitative conditions IHD can actually have greater power over QTL. It will be helpful to support this claim by providing some simulation results.

This comment prompted us to do a more detailed comparison of IHD vs. QTL power under conditions that are more similar to those observed in the BXD cohort. While preparing the original manuscript, we assumed that IHD might have greater power than QTL mapping in a population like the BXDs because some recombinant inbred lines have accumulated many more germline mutations than others (see Figure 1 in Sasani et al. 2022, Nature). In a quantitative trait locus scan (say, for the fraction of C>A mutations in each line) each BXD's mutation data would be weighted equally, even if a variable number of mutations was used to generate the phenotype point estimate in each line.

To address this, we performed a new series of simulations in which the average number of mutations per haplotype was allowed to vary. At the low end, some BXDs accumulated as few as 100 total germline mutations, while others have accumulated as many as 2,000. Thus, instead of simulating a mean number of mutations on each simulated haplotype, we allowed the mean number of mutations per haplotype to vary from N to 20N. By simulating a variable count of mutations on each haplotype, we could more easily test the benefits of comparing aggregate, rather than individual, mutation spectra between BXDs.

In these updated simulations, we find that IHD routinely outperforms QTL mapping under a range of parameter choices (see Author Response image 1). Since IHD aggregates the mutation spectra of all haplotypes with either B or D alleles at each locus in the genome, the method is much less sensitive to individual haplotypes with low mutation counts. We include a mention of these updated simulations on lines 135-138 and describe the updated simulations in greater detail in the Materials and Methods (lines 705-715).

**Author response image 1. sa4fig1:** Power of IHD and QTL mapping on simulated haplotypes with variable counts of mutations. We simulated germline mutations on the specified number of haplotypes (as described in the manuscript) but allowed the total number of mutations per haplotype to vary by a factor of 20.

1. The flip side of this advantage of IHD is that, when a significant association is detected, it is not immediately clear which mutation type is driving the signal. Related to this, it is unclear how the authors reached the point that "...the C>A mutator phenotype associated with the locus on chromosome 6", when they only detected significant IHD signal at rs46276051 (on Chr6), when conditioning on D genotypes at the rs27509845 (on Chr4) and no significant signal for any 1-mer mutation type by traditional mapping. The authors need to explain how they deduced that C>A mutation is the major source of the signal. In addition, beyond C>A mutations, can mutation types other than C>A contribute to the IHD signal at rs46276051? More generally, I hope the authors can provide some guidelines on how to narrow a significant IHD signal to specific candidate mutation type(s) affected, which will make the method more useful to other researchers.

We thank the reviewer for pointing out this gap in our logic. We omitted specific instructions for narrowing down an IHD signal to specific mutation type(s) for a few reasons. First, this can be addressed using mutational signature analysis methods that are in widespread use. For example, upon identifying one or more candidate mutator loci, we can enter the mutation spectra of samples with each possible mutator genotype into a program (e.g.,SigProfilerExtractor) to determine which combinations of mutation types occur proportionally more often in the genomes that harbor mutators (see Figure 3c in our manuscript). A second approach for narrowing down an IHD signal, highlighted in Figure 3a (and now described in the text of the Results section at lines 256-261), is to simply test which mutation type proportion(s) differ significantly between groups of samples with and without a candidate mutator (for example, with a Chi-square test of independence for each mutation type).

Although this second approach incurs a multiple testing burden, the burden is offset somewhat by using IHD to identify mutator loci, rather than performing association tests for every possible mutation type to begin with. Although Figure 3a only shows the significant difference in C>A fraction among BXDs with different mutator locus genotypes, Figure 3-figure supplement 1 shows the complete set of 1-mer spectrum comparisons. It is possible that this second approach would not prove very useful in the case of a mutator with a “flat” signature (i.e., a mutator that slightly perturbs the rates of many different mutation types), but in our case it clearly shows which mutation type is affected.

1. To account for differential relatedness between the inbred lines, the authors regressed the cosine distance between the two aggregate mutation spectra on the genome-wide genetic similarity and took the residual as the adjusted test metric. What is the value of the slope from this regression? If significantly non-zero, this would support a polygenic architecture of the mutation spectrum phenotype, which could be interesting. If not, is this adjustment really necessary? In addition, is the intercept assumed to be zero for this regression, and does such an assumption matter? I would appreciate seeing a supplemental figure on this regression.

The reviewer raises a good question. We find that the slope of the "distance vs. genetic similarity" regression is significantly non-zero, though the slope estimate itself is small. A plot of cosine distance vs. genome-wide genetic similarity (using all BXDs) is shown below in Author response image 2:

**Author response image 2. sa4fig2:** Relationship between cosine distance and genetic similarity in the BXDs. As described in the Materials and Methods, we computed two values at each marker in the BXDs: (1) the cosine distance between the aggregate mutation spectra of BXDs with either B or D genotypes at the marker, and (2) the correlation between genome-wide D allele frequencies in BXDs with either B or D genotypes at the marker. We then regressed these two values across all genome-wide markers.

This result indicates that if two groups of BXDs (one with D genotypes and one with B genotypes at a given locus) are more genetically similar, their mutation spectra are also more similar. Since the regression slope estimate is significantly non-zero (p < 2.2e-16), we believe that it's still worth using residuals as opposed to raw cosine distance values. This result also suggests that there may be a polygenic effect on the mutation spectrum in the BXDs.

We have also generated a plot showing the cosine distance between the mutation spectra of every possible pair of BXDs, regressed against the genetic similarity between each of those pairs (Author Response image 3). Here, the potential polygenic effects on mutation spectra similarity are perhaps more obvious.

**Author response image 3. sa4fig3:** Pairwise cosine distance between BXD mutation spectra as a function of genetic similarity. We computed two values for every possible pair of n = 117 BXDs: (1) the cosine distance between the samples' individual 1-mer mutation spectra and (2) the correlation coefficient between the samples' genome-wide counts of D alleles.

Private Comments1. It will also be useful to see how the power of IHD and QTL mapping depend on the allele frequency of the mutator allele and the sample size, as mutator alleles are likely rare or semi-rare in natural populations (such as the human de novo mutation dataset that the authors mentioned).

This is another good suggestion. In general, we'd expect the power of both IHD and QTL mapping to decrease as a function of mutator allele frequency. At the same time, we note that the power of these scans should mostly depend on the absolute number of carriers of the mutator allele and less on its frequency. In the BXD mouse study design, we observe high frequency mutators but also a relatively small sample size of just over 100 individuals. In natural human populations, mutator frequencies might be orders of magnitude smaller, but sample sizes may be orders of magnitude larger, especially as new cohorts of human genomes are routinely being sequenced. So, we expect to have similar power to detect a mutator segregating at, say, 0.5% frequency in a cohort of 20,000 individuals, as we would to detect a mutator segregating at 50% frequency in a dataset of 200 individuals.

To more formally address the reviewer's concern, we performed a series of simulations in which we simulated a population of 100 haplotypes. We assigned the same average number of mutations to each haplotype but allowed the allele frequency of the mutator allele to vary between 0.1, 0.25, and 0.5. The results of these simulations are shown in Author response image 4 and reveal that AMSD tends to have greater power than QTL mapping at lower mutator allele frequencies. We now mention these simulations in the text at lines 135-138 and include the simulation results in Figure 1-figure supplement 4.

**Author response image 4. sa4fig4:** Power of AMSD and QTL mapping on simulated haplotypes with variable marker allele frequencies. We simulated germline mutations on the specified number of haplotypes (as described in the manuscript), but simulated genotypes at the mutator allele such that "A" alleles were at the specified allele frequency.

1. In the Methods section of "testing for epistasis between the two mutator loci", it will be helpful to explicitly lay out the model and assumptions in mathematical formulae, in addition to the R scripts. For example, are the two loci considered independent when their effects on mutation rate is multiplicative or additive? Given the R scripts provided, it seems that the two loci are assumed to have multiplicative effects on the mutation rate, and that the mutation count follows a Poisson distribution with mean being the mutation rate times ADJ_AGE (i.e., the mutation opportunity times the number of generations of an inbred line). However, this is not easily understandable for readers who are not familiar with R language. In addition, I hope the authors can be more specific when discussing the epistatic interaction between the two loci by explicitly saying "synergistic effects beyond multiplicative effects on the C>A mutation rate".

The reviewer raises a good point about the clarity of our descriptions of tests for epistasis. We have now added a more detailed description of these tests in the section of the Materials and Methods beginning at line 875. We have also added a statement to the text at lines 289-291: “the combined effects of D genotypes at both loci exceed the sum of marginal effects of D genotypes at either locus alone.” We hope that this will help clarify the results of our tests for statistical epistasis.

**Reviewer 2 (Public Review):**
1. The main limitation of the approach is that it is difficult to see how it might be applied beyond the context of mutation accumulation experiments using recombinant inbred lines. This is because the signal it detects, and hence its power, is based on the number of extra accumulated mutations linked to (i.e. on the same chromosome as) the mutator allele. In germline mutation studies of wild populations the number of generations involved (and hence the total number of mutations) is typically small, or else the mutator allele becomes unlinked from the mutations it has caused (due to recombination), or is lost from the population altogether (due to chance or perhaps selection against its deleterious consequences).

The reviewer is correct that as it currently exists, IHD is mostly limited to applications in recombinant inbred lines (RILs) like the BXDs. This is due to the fact that IHD assumes that each diploid sample harbors one of two possible genotypes at a particular locus and ignores the possibility of heterozygous genotypes for simplicity. In natural, outbreeding populations, this assumption will obviously not hold. However, as we plan to further iterate on and improve the IHD method, we hope that it will be applicable to a wider variety of experimental systems in the future. We have added additional caveats about the applicability of our method to other systems in the text at lines 545-550.

Private Comments1. On p. 8, perhaps I've misunderstood but it's not clear in what way the SVs identified were relevant to the samples used in this dataset - were the founder strains assembled? Is there any chance that additional SVs were present, e.g. de novo early in the accumulation line?

Our description of this structural variation resource could have been clearer. The referenced SVs were identified in Ferraj et al. (2023) by generating high-quality long read assemblies of inbred laboratory mice. Both DBA/2J and C57BL/6J (the founder strains for the BXD resource) were included in the Ferraj et al. SV callset. We have clarified our description of the callset at lines 247-248.

It is certainly possible that individual BXD lines have accumulated de novo structural variants during inbreeding. However, these "private" SVs are unlikely to produce a strong IHD association signal (via linkage to one of the ~7,000 markers) at either the chromosome 4 or chromosome 6 locus, since we only tested markers that were at approximately 50% D allele frequency among the BXDs.

1. On p. 13, comparing the IHD and QTL approaches, regarding the advantage of the former in that it detects the combined effect of multiple k-mer mutation types, would it not be straightforward to aggregate counts for different types in a QTL setting as well?

The mutation spectrum is a multi-dimensional phenotype (6-dimensional if using the 1-mer spectrum, 96-dimensional if using the 3-mer spectrum, etc.). Most QTL mapping methods use linear models to test for associations between genotypes and a 1-dimensional phenotype (e.g., body weight, litter size). In the past, we used QTL mapping to test for associations between genotypes and a single element of the mutation spectrum (e.g., the rate of C>A mutations), but there isn't a straightforward way to aggregate or collapse the mutation spectrum into a 1dimensional phenotype that retains the information contained within the full 1-mer or 3-mer spectrum. For that reason, we developed the "aggregate mutation spectrum" approach, as it preserves information about the complete mutation spectrum in each group of strains.

The reviewer is correct that we could also aggregate counts of different mutation types to, say, perform a QTL scan for the load of a specific mutational signature. For example, we could first perform standard mutational signature analysis on our dataset and then test for QTLs associated with each signature that is discovered. However, this approach would not solve the second problem that our method is designed to solve: the appropriate weighting of samples based on how many mutations they contain.

1. pp. 15-16: In the discussion of how you account for relatedness between strains, I found the second explanation (on p. 16) much clearer. It would be interesting to know how much variance was typically accounted for by this regression?

As shown in the response to Reviewer 1, genotype similarity between genotype groups (i.e., those with either D or B genotypes at a marker) generally explains a small amount of variance in the cosine distance between those groups (R2~ = 0.007). However, since the slope term in that regression is significantly non-zero, correcting for this relationship should still improve our power relative to using raw cosine distance values that are slightly confounded by this relationship.

1. Similarly, in the section on Applying the IHD method to the BXDs (pp. 18-19), I think this description was very useful, and some or all of this description of the experiment (and how the DNMs in it arise) could profitably be moved to the introduction.

We appreciate the reviewer’s feedback about the details of the BXD cohort. Overall, we feel the description of the BXDs in the Introduction (at lines 65-73) is sufficient to introduce the cohort, though we now add some additional detail about variability in BXD inbreeding duration (at lines 89-93) to the Introduction as well, since it is quite relevant to some of the new simulation results presented in the manuscript.

1. A really minor one, not sure if this is for the journal or the authors, but it would be much better to include both page and line numbers in any version of an article for review. My pdf had neither!

We apologize for the lack of page/line numbers in the submitted PDF. We have now added line numbers to the revised version of the manuscript.

**Reviewer 3 (Public Review):**
1. Under simulated scenarios, the authors' new IHD method is not appreciably more powerful than conventional QTL mapping methods. While this does not diminish the rigor or novelty of the authors findings, it does temper enthusiasm for the IHD method's potential to uncover new mutators in other populations or datasets. Further, adaptation of this methodology to other datasets, including human trios or multigenerational families, will require some modification, which could present a barrier to broader community uptake. Notably, BXD mice are (mostly) inbred, justifying the authors consideration of just two genotype states at each locus, but this decision prevents out-of-the-box application to outbred populations and human genomic datasets. Lastly, some details of the IHD method are not clearly spelled out in the paper. In particular, it is unclear whether differences in BXD strain relatedness due to the breeding epoch structure are fully accounted for in permutations. The method's name - inter-haplotype distance - is also somewhat misleading, as it seems to imply that de novo mutations are aggregated at the scale of sub-chromosomal haplotype blocks, rather than across the whole genome.

The reviewer raises very fair concerns. As mentioned in response to a question from Reviewer 1, we performed additional simulation experiments that demonstrate the improved power of IHD (as compared to QTL mapping) in situations where mutation counts are variable across haplotypes or when mutator alleles are present at allele frequencies <50% (see Author response image 2 and 3, as well as new supplements to Figure 1 in the manuscript). However, the reviewer is correct that the IHD method is not applicable to collections of outbred individuals (that is, individuals with both heterozygous and homozygous genotypes), which will limit its current applications to datasets other than recombinant inbred lines. We have added a mention of these limitations to the Results at lines 138-141 and the Discussion at lines 545-550, but plan to iterate on the IHD method and introduce new features that enable its application to other datasets. We have also explicitly stated that we account for breeding epochs in our permutation tests in the Materials and Methods at lines 670-671. Both Reviewer 1 and Reviewer 3 raised concerns about the name of our method, and we have therefore changed “inter-haplotype distance” to “aggregate mutation spectrum distance” throughout the manuscript.

1. Nominating candidates within the chr6 mutator locus requires an approach for defining a credible interval and excluding/including specific genes within that interval as candidates. Sasani et al. delimit their focal window to 5Mb on either side of the SNP with the most extreme P-value in their IHD scan. This strategy suffers from several weaknesses. First, no justification for using 10 Mb window, as opposed to, e.g., a 5 Mb window or a window size delimited by a specific threshold of P-value drop, is given, rendering the approach rather ad hoc. Second, within their focal 10Mb window, the authors prioritize genes with annotated functions in DNA repair that harbor protein coding variants between the B6 and D2 founder strains. While the logic for focusing on known DNA repair genes is sensible, this locus also houses an appreciable number of genes that are not functionally annotated, but could, conceivably, perform relevant biological roles. These genes should not be excluded outright, especially if they are expressed in the germline. Further, the vast majority of functional SNPs are non-coding, (including the likely causal variant at the chr4 mutator previously identified in the BXD population). Thus, the author's decision to focus most heavily on coding variants is not well-justified. Sasani et al. dedicate considerable speculation in the manuscript to the likely identity of the causal variant, ultimately favoring the conclusion that the causal variant is a predicted deleterious missense variant in Mbd4. However, using a 5Mb window centered on the peak IHD scan SNP, rather than a 10Mb window, Mbd4 would be excluded. Further, SNP functional prediction accuracy is modest [e.g., PMID 28511696], and exclusion of the missense variant in Ogg1 due its benign prediction is potentially premature, especially given the wealth of functional data implicating Ogg1 in C>A mutations in house mice. Finally, the DNA repair gene closest to the peak IHD SNP is Rad18, which the authors largely exclude as a candidate.

We agree that the use of a 10 Mb window, rather than an empirically derived confidence interval, is a bit arbitrary and ad hoc. To address this concern, we have implemented a bootstrap resampling approach (Visscher et al. 1996, Genetics) to define confidence intervals surrounding IHD peaks. We have added a description of the approach to the Materials and Methods at lines 609-622, but a brief description follows. In each of N trials (here, N = 10,000), we take a bootstrap sample of the BXD phenotype and genotype data with replacement. We then perform an IHD scan on the chromosome of interest using the bootstrap sample and record the position of the marker with the largest cosine distance value (i.e., the "peak" marker). After N trials, we calculate the 90% confidence interval of bootstrapped peak marker locations; in other words, we identify the locations of two genotyped markers, between which 90% of all bootstrap trials produced an IHD peak. We note that bootstrap confidence intervals can exhibit poor "coverage" (a measure of how often the confidence intervals include the "true" QTL location) in QTL mapping studies (see Manichaikul et al. 2006, Genetics), but feel that the bootstrap is more reasonable than simply defining an ad hoc interval around an IHD peak.

The new 90% confidence interval surrounding the IHD peak on chromosome 6 is larger than the original (ad hoc) 10 Mbp window, now extending from around 95 Mbp to 114 Mbp. Notably, the new empirical confidence interval excludes Mbd4. We have accordingly updated our Results and Discussion sections to acknowledge the fact that Mbd4 no longer resides within the confidence interval surrounding the IHD peak on chromosome 6 and have added additional descriptions of genes that are now implicated by the 90% confidence interval. Given the uncertainties associated with using bootstrap confidence intervals, we have retained a brief discussion of the evidence supporting Mbd4 in the Discussion but focus primarily on Ogg1 as the most plausible candidate.

The reviewer raises a valid concern about our treatment of non-DNA repair genes within the interval surrounding the peak on chromosome 6. We have added more careful language to the text at lines 219-223 to acknowledge the fact that non-annotated genes in the confidence interval surrounding the chromosome 6 peak may play a role in the epistatic interaction we observed.

The reviewer also raises a reasonable concern about our discussions of both Mbd4 and Ogg1 as candidate genes in the Discussion. Since Mbd4 does not reside within the new empirical bootstrap confidence interval on chromosome 6 and given the strong prior evidence that Ogg1 is involved in C>A mutator phenotypes (and is in the same gene network as Mutyh), we have reframed the Discussion to focus on Ogg1 as the most plausible candidate gene (see lines 357360).

Using the GeneNetwork resource, we also more carefully explored the potential effects of noncoding variants on the C>A mutator phenotype we observed on chromosome 6. We have updated the Results at lines 240-246 and the Discussion at line 439-447 to provide more evidence for regulatory variants that may contribute to the C>A mutator phenotype. Specifically, we discovered a number of strong-effect cis-eQTLs for Ogg1 in a number of tissues, at which D genotypes are associated with decreased Ogg1 expression. Given new evidence that the original mutator locus we discovered on chromosome 4 harbors an intronic mobile element insertion that significantly affects Mutyh expression (see Ferraj et al. 2023, Cell Genomics), it is certainly possible that the mutator phenotype associated with genotypes on chromosome 6 may also be mediated by regulatory, rather than coding, variation.

1. Additionally, some claims in the paper are not well-supported by the author's data. For example, in the Discussion, the authors assert that "multiple mutator alleles have spontaneously arisen during the evolutionary history of inbred laboratory mice" and that "... mutational pressure can cause mutation rates to rise in just a few generations of relaxed selection in captivity". However, these statements are undercut by data in this paper and the authors' prior publication demonstrating that a number of candidate variants are segregating in natural mouse populations. These variants almost certainly did not emerge de novo in laboratory colonies, but were inherited from their wild mouse ancestors. Further, the wild mouse population genomic dataset used by the authors falls far short of comprehensively sampling wild mouse diversity; variants in laboratory populations could derive from unsampled wild populations.

The reviewer raises a good point. In our previous publication (Sasani et al. 2022, Nature), we hypothesized that Mutyh mutator alleles had arisen in wild, outbreeding populations of *Mus musculus*, and later became fixed in inbred strains like DBA/2J and C57BL/6J. However, in the current manuscript, we included a statement about mutator alleles "spontaneously arising during the evolutionary history of inbred laboratory mice" to reflect new evidence (from Ferraj et al. 2023, Cell Genomics) that the mutator allele we originally identified in Mutyh may not be wild derived after all. Instead, Ferraj et al. suggest that the C>A mutator phenotype we originally identified is caused by an intronic mobile element insertion (MEI) that is present in DBA/2J and a handful of other inbred laboratory strains. Although this MEI may have originally occurred in a wild population of mice, we wanted to acknowledge the possibility that both the original Mutyh mutator allele, as well as the new mutator allele(s) we discovered in this manuscript, could have arisen during the production and inbreeding of inbred laboratory lines. We have also added language to the Discussion at lines 325-327 to acknowledge that the 67 wild mice we analyzed do not comprise a comprehensive picture of the genetic diversity present in wild-derived samples.

We have added additional language to the Discussion at lines 349-357 in which we acknowledge that the chromosome 6 mutator allele might have originated in either laboratory or wild mice and elaborate on the possibility that mutator alleles with deleterious fitness consequences may be more likely to persist in inbred laboratory colonies.

1. Finally, the implications of a discovering a mutator whose expression is potentially conditional on the genotype at a second locus are not raised in the Discussion. While not a weakness per se, this omission is perceived to be a missed opportunity to emphasize what, to this reviewer, is one of the most exciting impacts of this work. The potential background dependence of mutator expression could partially shelter it from the action of selection, allowing the allele persist in populations. This finding bears on theoretical models of mutation rate evolution and may have important implications for efforts to map additional mutator loci. It seems unfortunate to not elevate these points.

We agree and have added additional discussion of the possibility that the C>A mutator phenotypes in the BXDs are a result of interactions between the expression of two DNA repair genes in the same base-excision network to the Discussion section at lines 447-449.

Private comments1. The criteria used to determine or specify haplotype size are not specified in the manuscript. I mention this above but reiterate here as this was a big point of confusion for me when reading the paper. Haplotype length is important consideration for overall power and for proper extension of this method to other systems/populations.

We may not have been clear enough in our description of our method, and as suggested by Reviewer 1, the name "inter-haplotype distance" may also have been a source of confusion. At a given marker, we compute the aggregate mutation spectrum in BXDs with either B or D genotypes using all genome-wide de novo mutations observed in those BXDs. Since the BXDs were inbred for many generations, we expect that almost all de novo germline mutations observed in an RIL are in near-perfect linkage with the informative genotypes used for distance scans. Thus, the "haplotypes" used in the inter-haplotype distance scans are essentially the lengths of entire genomes.

1. Results, first paragraph, final sentence. I found the language here confusing. I don't understand how one can compute the cosine distance at single markers, as stated. I'm assuming cosine distance is computed from variants residing on haplotypes delimited by some defined window surrounding the focal marker?

As discussed above, we aggregate all genome-wide de novo mutations in each group of BXDs at a given marker, rather than only considering DNMs within a particular window surrounding the marker. The approach is discussed in greater detail in the caption of Figure 1.

1. Nominating candidates for the chr6 locus, Table 1. It would be worth confirming that the three prioritized candidates (Setmar, Ogg1, and Mbd4) all show germline expression.

Using the Mouse Genome Informatics online resource, we confirmed that all prioritized candidate genes (now including Setmar and Ogg1, but not Mbd4) are expressed in the male and female gonads, and mention this in the Results at lines 228 and 233-234.

1. Does the chr6 peak on the C>A LOD plot (Figure 2- figure supplement 1) overlap the same peak identified in the IHD scan? And, does this peak rise to significance when using alpha = 0.05? Given that the goal of these QTL scans is to identify loci that interact with the C>A mutator on chr4, it is reasonable to hypothesize that the mutation impact of epistatic loci will also be restricted to C>A mutations. Therefore, I am not fully convinced that the conservative alpha = 0.05/7 threshold is necessary.

The chromosome 6 peak in Figure 2-figure supplement 1 does, in fact, overlap the peak marker we identified on chromosome 6 using IHD. One reason we decided to use a more conservative alpha of (0.05 / 7) is that we wanted these results to be analogous to the ones we performed in a previous paper (Sasani et al. 2022, Nature), in which we first identified the mutator locus on chromosome 4. However, the C>A peak does not rise to genome-wide significance if we use a less conservative alpha value of 0.05 (see Author response image 5). As discussed in our response to Reviewer 1, we find that QTL mapping is not as powerful as IHD when haplotypes have accumulated variable numbers of germline mutations (as in the BXDs), which likely explains the fact that the peak on chromosome 6 is not genome-wide significant using QTL mapping.

**Author response image 5. sa4fig5:** QTL scan for the fraction of C>A mutations in BXDs harboring D alleles at the locus near MythQTL scan was performed at a genome-wide significance alpha of 0. 05, rather than 0.05/7.

1. Is there significant LD between the IHD peaks on chr6 and chr4 across the BXD? If so, it could suggest that the signal is driven by cryptic population structure that is not fully accounted for in the author's regression based approach. If not, this point may merit an explicit mention in the text as an additional validation for the authenticity of the chr6 mutator finding.

This is a good question. We used the

1. Discussion, last sentence of the "Possible causal alleles..." section: I don't understand how the absence of the Mariner-family domain leads the authors to this conclusion. Setmar is involved in NHEJ, which to my knowledge is not a repair process that is expected to have a specific C>A mutation bias. I think this is grounds enough for ruling out its potential contributions, in favor of focusing on other candidates, (e.g., Mbd4 and Ogg1).

The reviewer raises a good point. Our main reason for mentioning the absence of the Marinerfamily domain is that even if NHEJ were responsible for the C>A mutator phenotype, it likely wouldn't be possible for Setmar to participate in NHEJ without the domain. However, the reviewer is correct that NHEJ is not expected to cause a C>A mutation bias, and we have added a mention of this to the text as well at lines 379-382.

1. Discussion, second to last paragraph of section "Mbd4 may buffer...": The authors speculate that reduced activity of Mbd4 could modulate rates of apoptosis in response to DNA damage. This leads to the prediction that mice with mutator alleles at both Mutyh and Mbd4 should exhibit higher overall mutation rates compared to mice with other genotypes. This possibility could be tested with the authors' data.

The reviewer raises a good question. As mentioned above, however, we implemented a new approach to calculate confidence intervals surrounding distance peaks and found that this empirical approach (rather than the ad hoc 10-Mbp window approach we used previously) excluded Mbd4 from the credible interval. Although we still mention Mbd4 as a possible candidate (since it still resides within the 10 Mbp window), we have refactored the Discussion section to focus primarily on the evidence for Ogg1 as a candidate gene on chromosome 6.

In any case, we do not observe that mice with mutator alleles at both the chromosome 4 and chromosome 6 loci have higher overall mutation rates compared to mice with other genotype combinations. This may not be terribly surprising, however, since C>A mutations only comprise about 10% of all possible mutations. Thus, given the variance in other 1-mer mutation counts, even a substantial increase in the C>A mutation rate might not have a detectable effect on the overall mutation rate. Indeed, in our original paper describing the Mutyh mutator allele (Sasani et al. 2022, Nature), we did not identify any QTL for the overall mutation rate in the BXDs and found that mice with the chromosome 4 mutator allele only exhibited a 1.11X increase in their overall mutation rates relative to mice without the mutator allele.

1. Methods, "Accounting for BXD population structure": An "epoch-aware" permutation strategy is described here, but it is not clear when (and whether) this strategy is used to determine significance of IHD P-values.

We have added a more explicit mention of this to the Methods section at lines 670-671, as we do, in fact, use the epoch-aware permutation strategy when calculating empirical distance thresholds.

1. The simulation scheme employed for power calculations is highly specific to the BXD population. This is not a weakness, and perfectly appropriate to the study population used here. However, it does limit the transferability of the power analyses presented in this manuscript to other populations. This limitation may merit an explicit cautionary mention to readers who may aspire to port the IHD method over to their study system.

This is true. Our simulation strategy is relatively simple and makes a number of assumptions about the simulated population of haplotypes (allele frequencies normally distributed around 0.5, expected rates of each mutation type, etc.). In response to concerns from Reviewer 1, we performed an updated series of simulations in which we varied some of these parameters (mutator allele frequencies, mean numbers of mutations on haplotypes, etc.). However, we have added a mention of the simulation approach's limitations and specificity to the BXDs to the text at lines 545-550.